# Infant and adult SCA13 mutations differentially affect Purkinje cell excitability, maturation, and viability in vivo

Jui-Yi Hsieh[1,2,†], Brittany N Ulrich[1,2,†], Fadi A Issa[1], Meng-chin A Lin[1], Brandon Brown[1], Diane M Papazian[1,2,3,4]*

[1]Department of Physiology, David Geffen School of Medicine at UCLA, Los Angeles, United States; [2]Interdepartmental PhD Program in Molecular, Cellular, and Integrative Physiology, David Geffen School of Medicine at UCLA, Los Angeles, United States; [3]Brain Research Institute, UCLA, Los Angeles, United States; [4]Molecular Biology Institute, UCLA, Los Angeles, United States

**Abstract** Mutations in *KCNC3*, which encodes the Kv3.3 K$^+$ channel, cause spinocerebellar ataxia 13 (SCA13). SCA13 exists in distinct forms with onset in infancy or adulthood. Using zebrafish, we tested the hypothesis that infant- and adult-onset mutations differentially affect the excitability and viability of Purkinje cells in vivo during cerebellar development. An infant-onset mutation dramatically and transiently increased Purkinje cell excitability, stunted process extension, impaired dendritic branching and synaptogenesis, and caused rapid cell death during cerebellar development. Reducing excitability increased early Purkinje cell survival. In contrast, an adult-onset mutation did not significantly alter basal tonic firing in Purkinje cells, but reduced excitability during evoked high frequency spiking. Purkinje cells expressing the adult-onset mutation matured normally and did not degenerate during cerebellar development. Our results suggest that differential changes in the excitability of cerebellar neurons contribute to the distinct ages of onset and timing of cerebellar degeneration in infant- and adult-onset SCA13.

**\*For correspondence:**
papazian@mednet.ucla.edu

[†]These authors contributed equally to this work

**Competing interests:** The authors declare that no competing interests exist.

## Introduction

Spinocerebellar ataxias (SCAs) are a group of rare, autosomal dominant diseases characterized by locomotor deficits and cerebellar degeneration, typically with onset in adulthood (*Klockgether et al., 2019*). Spinocerebellar ataxia 13 (SCA13) is caused by mutations in the *KCNC3* gene, which encodes the Kv3.3 voltage-gated K$^+$ channel (*Waters et al., 2006*; *Figueroa et al., 2010*; *Figueroa et al., 2011*; *Duarri et al., 2015*; *Zhang and Kaczmarek, 2016*). Unlike most SCAs, SCA13 exists in two forms that differ in the age of onset and associated symptoms (*Herman-Bert et al., 2000*; *Waters et al., 2005*; *Waters et al., 2006*; *Figueroa et al., 2010*; *Figueroa et al., 2011*; *Duarri et al., 2015*). The early-onset form is evident in infancy with motor delay, persistent motor deficits, and intellectual disability (*Herman-Bert et al., 2000*). Severe cerebellar maldevelopment and atrophy have been observed by MR imaging by 10 months of age (*Khare et al., 2017*). In contrast, the adult-onset form typically emerges in the third or fourth decade of life with progressive ataxia accompanied by progressive degeneration of the cerebellum (*Subramony et al., 2013*). The infant- and adult-onset forms of SCA13 are caused by distinct mutations in Kv3.3 (*Waters et al., 2006*; *Figueroa et al., 2010*; *Figueroa et al., 2011*; *Duarri et al., 2015*). The age of onset is strongly correlated with the same mutation in unrelated families, indicating that the two forms of

SCA13 do not reflect differences in genetic background (*Waters et al., 2006*; *Figueroa et al., 2010*; *Figueroa et al., 2011*; *Duarri et al., 2015*).

To understand disease mechanisms in SCA13, it is essential to determine how different mutations in the same gene give rise to distinct clinical phenotypes. We have shown that infant- and adult-onset mutations have differential effects on Kv3.3 function that may underlie the two forms of the disease (*Waters et al., 2006*; *Minassian et al., 2012*). An adult-onset mutation, R420H, which changes the third arginine in the S4 transmembrane segment to histidine, does not generate functional channels when expressed alone (*Waters et al., 2006*; *Minassian et al., 2012*). Upon co-assembly with wild-type subunits in the tetrameric channel, the mutant subunit suppresses Kv3.3 activity by a dominant negative mechanism (*Minassian et al., 2012*). Under physiological conditions, the functional properties of the residual current do not differ significantly from wild type (*Minassian et al., 2012*). In contrast, several infant-onset mutations, including R423H, which changes the fourth arginine in S4 to histidine, have dominant gain-of-function effects on channel gating, with or without an accompanying dominant negative effect (*Waters et al., 2006*; *Minassian et al., 2012*; *Duarri et al., 2015*). This is significant because Kv3.3, like other Kv3 family members, has specialized gating properties that shape the functional repertoire of neurons (*Rudy and McBain, 2001*). Kv3.3 activates in a depolarized voltage range normally attained only during action potentials (*Rudy and McBain, 2001*). As a result, the channel does not contribute significantly to maintaining the resting potential or modulating excitability near threshold. During an action potential, Kv3.3 channels open with fast kinetics, leading to rapid repolarization, brief spikes, and efficient recovery of voltage-gated $Na^+$ channels from inactivation (*Rudy and McBain, 2001*). Upon repolarization, Kv3.3 channels close quickly, shortening the afterhyperpolarization and facilitating the next action potential (*Rudy and McBain, 2001*). These gating properties promote sustained, high frequency firing of action potentials in neurons (*Rudy and McBain, 2001*).

Kv3.3 is highly expressed in cerebellar Purkinje cells, where it contributes to the mechanism of spontaneous pacemaking (*Martina et al., 2003*; *Akemann and Knöpfel, 2006*). Kv3.3 is co-expressed in Purkinje cells with the $Na_v1.6$ voltage-gated $Na^+$ channel (*Raman et al., 1997*; *Khaliq et al., 2003*; *Martina et al., 2003*; *Akemann and Knöpfel, 2006*). During an action potential, $Na_v1.6$ is subject to open channel block conferred by an auxiliary subunit thought to be $Na_v\beta4$ or FGF14 (*Grieco et al., 2005*; *White et al., 2019*). Rapid repolarization mediated by Kv3.3 relieves open channel block of $Na^+$ channels, generating a resurgent $Na^+$ current in the interspike interval that triggers the next action potential (*Raman and Bean, 1997*; *Khaliq et al., 2003*; *Grieco et al., 2005*). Together, these channels regulate the spontaneous tonic firing that is characteristic of Purkinje cells (*Raman et al., 1997*; *Khaliq et al., 2003*; *Martina et al., 2003*; *Akemann and Knöpfel, 2006*).

The locomotor deficits in SCA13 are primarily cerebellar in origin (*Stevanin et al., 2005*; *Waters and Pulst, 2008*; *Subramony et al., 2013*; *Klockgether et al., 2019*). It is therefore important to determine the effects of disease-causing mutations in cerebellar neurons in vivo. Given the role of Kv3.3 in controlling Purkinje cell firing (*Akemann and Knöpfel, 2006*; *Martina et al., 2003*; *Martina et al., 2007*; *McMahon et al., 2004*), we tested the hypothesis that infant- and adult-onset mutations have differential effects on Purkinje cell excitability that are correlated with age-dependent changes in Purkinje cell viability. We focused on the R420H and R423H mutations—an intriguing pair for study because these arginine-to-histidine mutations are separated by only two amino acids in the S4 segment in the voltage sensor domain, but cause the distinct adult- and infant-onset forms of SCA13, respectively (*Figure 1A*; *Waters et al., 2006*; *Figueroa et al., 2010*; *Figueroa et al., 2011*). These mutations will be referred to as aR3H and iR4H because they change the third and fourth arginine residues in S4 to histidine; 'a' and 'i' denote their association with the adult- and infant-onset forms of the disease, respectively.

We generated the aR3H and iR4H mutations in zebrafish Kv3.3a and expressed them in Purkinje cells in the zebrafish cerebellum. Purkinje cell activity was recorded in living animals using a patch clamp (*Hsieh et al., 2014*; *Hsieh and Papazian, 2018*). In parallel, Purkinje cell maturation and survival were monitored by live confocal imaging during cerebellar development. We report that infant- and adult-onset SCA13 mutations have opposing effects on excitability. iR4H causes transient hyper-excitability. In contrast, aR3H does not affect spontaneous tonic firing under basal conditions, but reduces excitability during evoked, high-frequency firing. We further report that iR4H and aR3H have dramatically different effects on the morphological maturation and survival of Purkinje cells

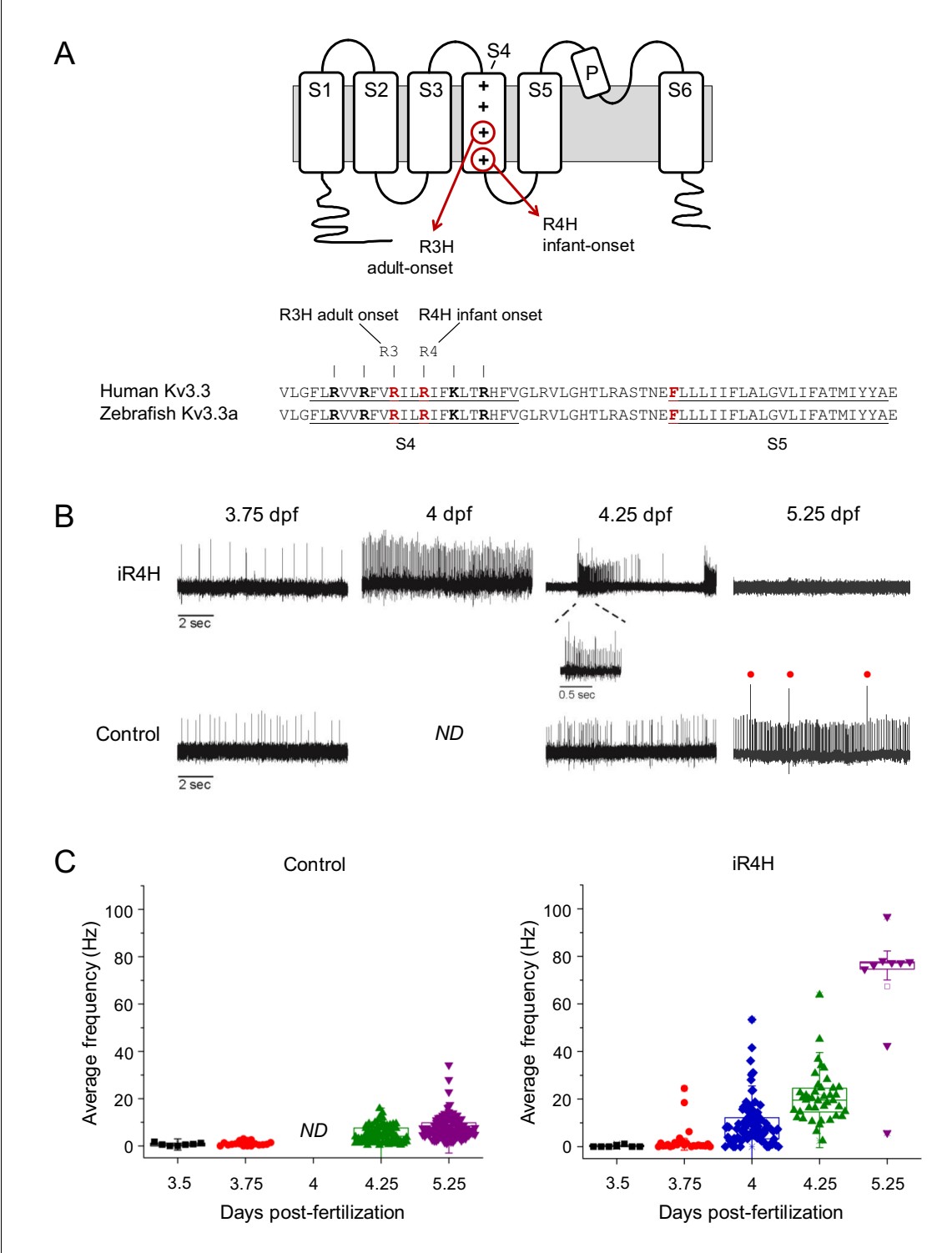

**Figure 1.** iR4H expression dramatically and transiently increases Purkinje cell excitability. (**A**) Top: cartoon of membrane topology of one Kv3.3 subunit shows transmembrane segments S1-S6, re-entrant P loop, and approximate locations of the adult-onset R3H and infant-onset R4H SCA13 mutations in S4. The plus signs in S4 represent the positively charged arginine residues that sense changes in voltage and initiate conformational changes that result in pore opening (*Seoh et al., 1996*). The aR3H mutation corresponds to R420H and R335H in human and zebrafish Kv3, respectively (*Mock et al., 2010*). The iR4H mutation corresponds to R423H and R338H in human and zebrafish Kv3, respectively (*Mock et al., 2010*). Bottom: alignment shows the identity of S4, S4-S5 linker, and S5 sequences from human Kv3.3 and zebrafish Kv3.3a. The underlined and bolded residues shown in red indicate the locations of aR3H and iR4H in S4, and the location of a phenylalanine residue at position 448 (F448) in human Kv3.3. The mutation F448L (iFL),

*Figure 1 continued on next page*

*Figure 1 continued*

mentioned in the text, causes infant-onset SCA13 (*Waters et al., 2006*). (**B**) Representative loose patch recordings of spontaneous firing in Purkinje cells at different times between 3.75 and 5.25 dpf, as indicated (*Hsieh et al., 2014*). Upper row: Purkinje cells expressing iR4H. Lower row: control Purkinje cells. A portion of the trace recorded from an iR4H-expressing cell at 4.25 dpf is shown on an expanded time scale to illustrate an interval of high frequency firing. Red dots: complex spikes. ND: not done. (**C**) Average frequency of simple spikes was calculated from individual 10 s traces and plotted versus the recording time post-fertilization. Each symbol represents a trace. For control cells, mean firing frequencies ± SEM were 0.7 ± 0.2, 1.0 ± 0.2, 5.1 ± 0.4, and 7.7 ± 0.5 Hz at 3.5, 3.75, 4.25, and 5.25 dpf, respectively (*n* = 7–117). No recordings from control cells were made at the 4 dpf time point. For iR4H-expressing cells, mean firing frequencies ± SEM were 0.3 ± 0.2, 2.7 ± 1.2, 9.7 ± 1.0, 20.8 ± 1.7, and 67.4 ± 9.0 Hz at 3.5, 3.75, 4, 4.25, and 5.25 dpf, respectively (*n* = 7–92). Note that most iR4H-expressing Purkinje cells were electrically silent at 5.25 dpf. Data are shown only for those cells that fired at that time point. Average frequency of simple spikes differed significantly between control and iR4H-expressing Purkinje cells at 4.25 (p=8.3×10$^{-21}$) and 5.25 dpf (p=3.7×10$^{-42}$) by two-sided unpaired t-test. Histograms that compare firing frequencies in control and iR4H-expressing cells are provided in *Figure 1—figure supplement 2A*. The fold-change in firing frequencies of iR4H-expressing cells compared to control cells is shown in *Figure 1—figure supplement 2B*. Average firing frequencies in control cells are shown on an expanded frequency scale in *Figure 1—figure supplement 3*.

The online version of this article includes the following figure supplement(s) for figure 1:

**Figure supplement 1.** Functional effects of iR4H mutation are conserved in zebrafish Kv3.3.

**Figure supplement 2.** Expression of iR4H in Purkinje cells results in hyperexcitability.

**Figure supplement 3.** Average frequency of simple spikes in control Purkinje cells is shown as a function of time postfertilization.

during cerebellar development. The infant-onset mutation stunts process extension, impairs synaptogenesis and dendritic branching, and results in rapid Purkinje cell death. Suppressing hyperexcitability promotes the early survival of iR4H-expressing neurons, suggesting that increased excitability contributes to Purkinje cell degeneration. In contrast, aR3H-expressing Purkinje cells mature normally and survive robustly through cerebellar development. The distinct effects of the iR4H and aR3H mutations on Purkinje cell maturation and survival during cerebellar development in vivo are reminiscent of the age-dependent cerebellar degeneration seen in infant- and adult-onset SCA13 (*Herman-Bert et al., 2000*; *Waters et al., 2006*; *Khare et al., 2017*; *Waters and Pulst, 2008*; *Subramony et al., 2013*). We propose that the defective maturation and highly penetrant death of iR4H-expressing Purkinje cells in zebrafish during cerebellar development are mechanistically related to the cerebellar maldevelopment, atrophy, and degeneration that occurs during infancy in patients with early-onset SCA13. Our results suggest that hyperexcitability in Purkinje cells may contribute to maldevelopment and atrophy of the cerebellum in infant-onset SCA13.

## Results

### Infant-onset iR4H mutation transiently increases Purkinje cell excitability during cerebellar development in vivo

We have previously shown that zebrafish Purkinje cells, which are born on the 3$^{rd}$ day post-fertilization (*Bae et al., 2009*), become spontaneously active within hours, coinciding with the expression of Kv3.3 and Na$_v$1.6, ion channels that regulate spontaneous pacemaking activity in mammalian Purkinje cells (*Hsieh et al., 2014*; *Raman and Bean, 1997*; *Khaliq et al., 2003*; *Martina et al., 2003*; *Akemann and Knöpfel, 2006*). The frequency and regularity of tonic firing increase until ~5 days post-fertilization (dpf) and then remain stable through at least 14 dpf (*Hsieh et al., 2014*). By 5 dpf, most cells also exhibit complex spiking in response to synaptic input from climbing fibers originating in the inferior olive (*Hsieh et al., 2014*). The properties of tonic action potential firing and complex spiking in Purkinje cells are similar in zebrafish and mammals (*Hsieh et al., 2014*; *Sengupta and Thirumalai, 2015*; *Scalise et al., 2016*; *Harmon et al., 2017*).

To test the hypothesis that infant- and adult-onset SCA13 mutations alter Purkinje cell excitability in distinct ways, the aR3H or iR4H mutant subunits or exogenous wild-type Kv3.3 subunits were expressed in the zebrafish cerebellum under the control of the Purkinje cell-specific *aldoca* promoter using F0 transgenesis, which results in a mosaic pattern of expression (*Kikuta and Kawakami, 2009*; *Tanabe et al., 2010*). Expressing cells were identified by the presence of mCherry, which was produced as a separate protein from the same plasmid using a 2A sequence (*Kim et al., 2011*). Neuronal excitability was characterized by recording action potential firing using extracellular electrodes in the loose patch configuration (*Hsieh et al., 2014*; *Hsieh and Papazian, 2018*). Experiments were

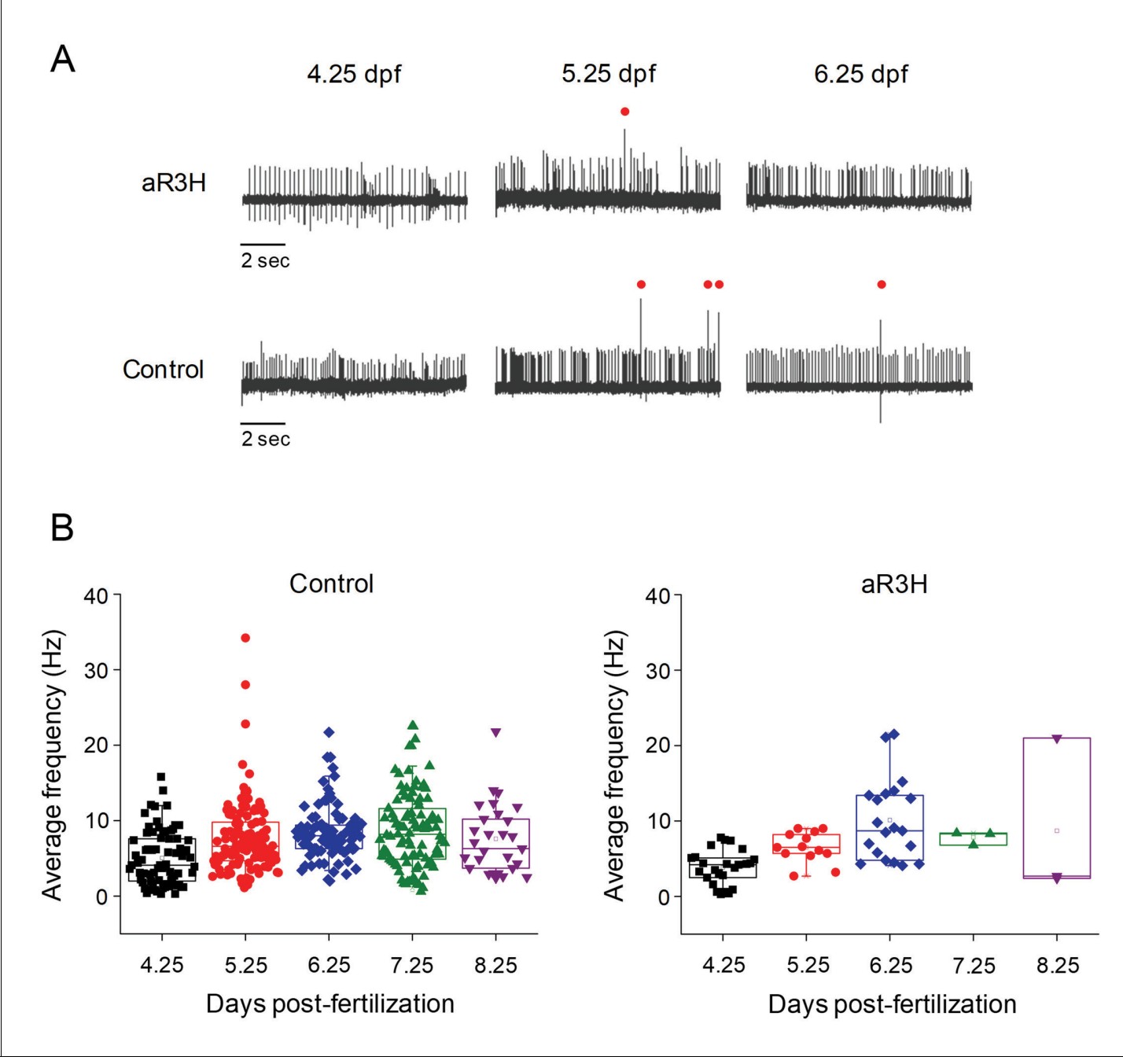

**Figure 2.** aR3H expression does not significantly alter Purkinje cell excitability under basal conditions. (**A**) Representative loose patch recordings of spontaneous firing in Purkinje cells at 4.25, 5.25, and 6.25 dpf, as indicated (*Hsieh et al., 2014*). Upper row: Purkinje cells expressing aR3H. Lower row: control Purkinje cells. Red dots: complex spikes. (**B**) Average frequency of simple spikes was calculated from individual 10 s traces and plotted versus the recording time post-fertilization between 4.25 and 8.25 dpf, as indicated. Each symbol represents a trace. For control cells, mean firing frequencies ± SEM were 5.1 ± 0.4, 7.7 ± 0.5, 8.5 ± 0.4, 8.7 ± 0.5, and 7.6 ± 0.9 Hz at 4.25, 5.25, 6.25, 7.25, and 8.25 dpf, respectively ($n$ = 29–117). Control data obtained at 4.25 and 5.25 dpf are the same as shown in *Figure 1C* and are repeated here for comparison to aR3H data. For aR3H-expressing cells, mean firing frequencies ± SEM were 3.8 ± 0.5, 6.5 ± 0.6, 10.1 ± 1.2, 7.8 ± 0.5, and 8.7 ± 6.2 Hz at 4.25, 5.25, 6.25, 7.25, and 8.25 dpf, respectively ($n$ = 3–25). Values in control and aR3H-expressing cells did not differ significantly. Histograms that compare firing frequencies in control and aR3H-expressing cells are provided in *Figure 2—figure supplement 1*. The regularity of tonic firing was assessed by calculating CV2. The mean values ± SEM were 0.77 ± 0.04, 0.58 ± 0.02, 0.47 ± 0.02, 0.53 ± 0.02, and 0.63 ± 0.03 for control cells ($n$ = 29–117) and 0.58 ± 0.06, 0.47 ± 0.12, 0.23 ± 0.05, 0.61 ± 0.2, and 0.52 for aR3H-expressing Purkinje cells ($n$ = 2–25) at 4.25, 5.25, 6.25, 7.25, and 8.25 dpf, respectively. CV2 values in control versus aR3H cells differed significantly at 4.25 dpf (p=9.5×10$^{-3}$) and 6.25 dpf (p=6.1×10$^{-7}$) but not at other time points (p>0.05) by two-sided unpaired t-test.

The online version of this article includes the following figure supplement(s) for figure 2:

*Figure 2 continued on next page*

*Figure 2 continued*

**Figure supplement 1.** Expression of aR3H in Purkinje cells does not alter basal excitability.

**Figure supplement 2.** Average frequency of complex spiking is not significantly altered by expression of aR3H.

**Figure supplement 3.** Expression of exogenous wild-type Kv3.3 (exoWT) does not significantly alter Purkinje cell excitability.

performed in live, awake zebrafish. Results were compared to Purkinje cells in a transgenic zebrafish line that expresses a membrane-bound form of the Venus yellow fluorescent protein under the control of the *aldoca* promotor (*Hsieh et al., 2014*).

It is important to note that the iR4H and aR3H subunits are non-functional in the absence of wild-type Kv3 channel subunits. This has been demonstrated for the aR3H mutation introduced into either mammalian or zebrafish Kv3.3 by expression and voltage clamp analysis in *Xenopus* oocytes (*Waters et al., 2006*; *Minassian et al., 2012*; *Mock et al., 2010*). It has also been shown for the iR4H mutation introduced into mammalian Kv3.3 (*Figueroa et al., 2010*; *Minassian et al., 2012*), a finding that we have now confirmed for iR4H in zebrafish Kv3.3 (*Figure 1—figure supplement 1*). Accordingly, these mutant subunits would have to co-assemble with endogenous, wild-type Kv3 subunits to affect Purkinje cell excitability.

Expression of the iR4H mutation dramatically increased Purkinje cell excitability soon after the emergence of spontaneous tonic firing at ~3.5 dpf (*Figure 1B,C*). By 3.75 dpf, a few iR4H-expressing neurons were firing at an unusually high frequency (*Figure 1C*; *Figure 1—figure supplement 2*). The fraction of iR4H-expressing cells that were hyperexcitable increased dramatically by 4 dpf (*Figure 1C*; *Figure 1—figure supplement 2*). By 4.25 dpf, there was relatively little overlap in the range of average firing frequencies seen in control and iR4H-expressing cells (*Figure 1C*; *Figure 1—figure supplements 2* and *3*). Although a substantial fraction of iR4H-expressing cells exhibited sustained tonic firing at 4 dpf, by 4.25 dpf, most fired intermittently, with high frequency bouts of action potential spiking separated by intervals of reduced activity (*Figure 1B*).

The hyperexcitability of iR4H-expressing Purkinje cells was transient. By 5.25 dpf, most cells expressing iR4H had fallen silent (*Figure 1B*). This was not observed in control cells, the vast majority of which fired robustly by 4.25 dpf and at subsequent times. In contrast, between 5 and 7 dpf, fewer than 5–10% of iR4H-expressing cells fired action potentials. The rest were completely inactive. Those that continued to fire did so at abnormally high frequencies (*Figure 1C*; *Figure 1—figure supplement 2*).

In control Purkinje cells, functional climbing fiber input from inferior olive neurons was first observed at 3.75 dpf when a minority of cells (~17%) fired complex spikes. This increased to ~37% by 4.25 dpf and ~70% by 5.25 dpf (*Figure 1B*; *Hsieh et al., 2014*). In contrast, no complex spikes were recorded from a comparable number of iR4H-expressing cells at 3.75 dpf, and only ~17% fired complex spikes at 4.25 dpf. No complex spiking was observed at 5.25 dpf, in accord with the observation that most iR4H-expressing Purkinje cells were electrically silent by that time.

## Adult-onset aR3H mutation decreases Purkinje cell excitability during evoked high frequency firing

In contrast to iR4H, expression of the adult-onset aR3H mutation did not significantly alter the excitability of Purkinje cells under basal conditions (*Figure 2*; *Figure 2—figure supplement 1*). The average frequency of spontaneous tonic firing did not differ significantly in aR3H-expressing and control cells between 4.25 and 8.25 dpf (*Figure 2A,B*). Complex spiking emerged normally in aR3H-expressing neurons, with no significant difference in frequency compared to control cells (*Figure 2—figure supplement 2*). Similarly, expression of exogenous wild-type Kv3.3 did not alter the frequencies of tonic firing or complex spiking in Purkinje cells compared to controls (*Figure 2—figure supplement 3*).

Kv3 channels promote sustained high frequency firing in neurons by maintaining the availability of voltage-gated Na$^+$ channels (*Rudy and McBain, 2001*). Previous work indicates that reducing Kv3 activity with channel blockers or by genetic deletion of *Kcnc* genes in mice impairs sustained fast spiking due to the increased accumulation of Na$^+$ channel inactivation (*Erisir et al., 1999*; *Lau et al., 2000*; *Akemann and Knöpfel, 2006*; *Rudy and McBain, 2001*). We have shown that expression of the dominant negative aR3H subunit significantly decreases the excitability of Kv3.3-expressing

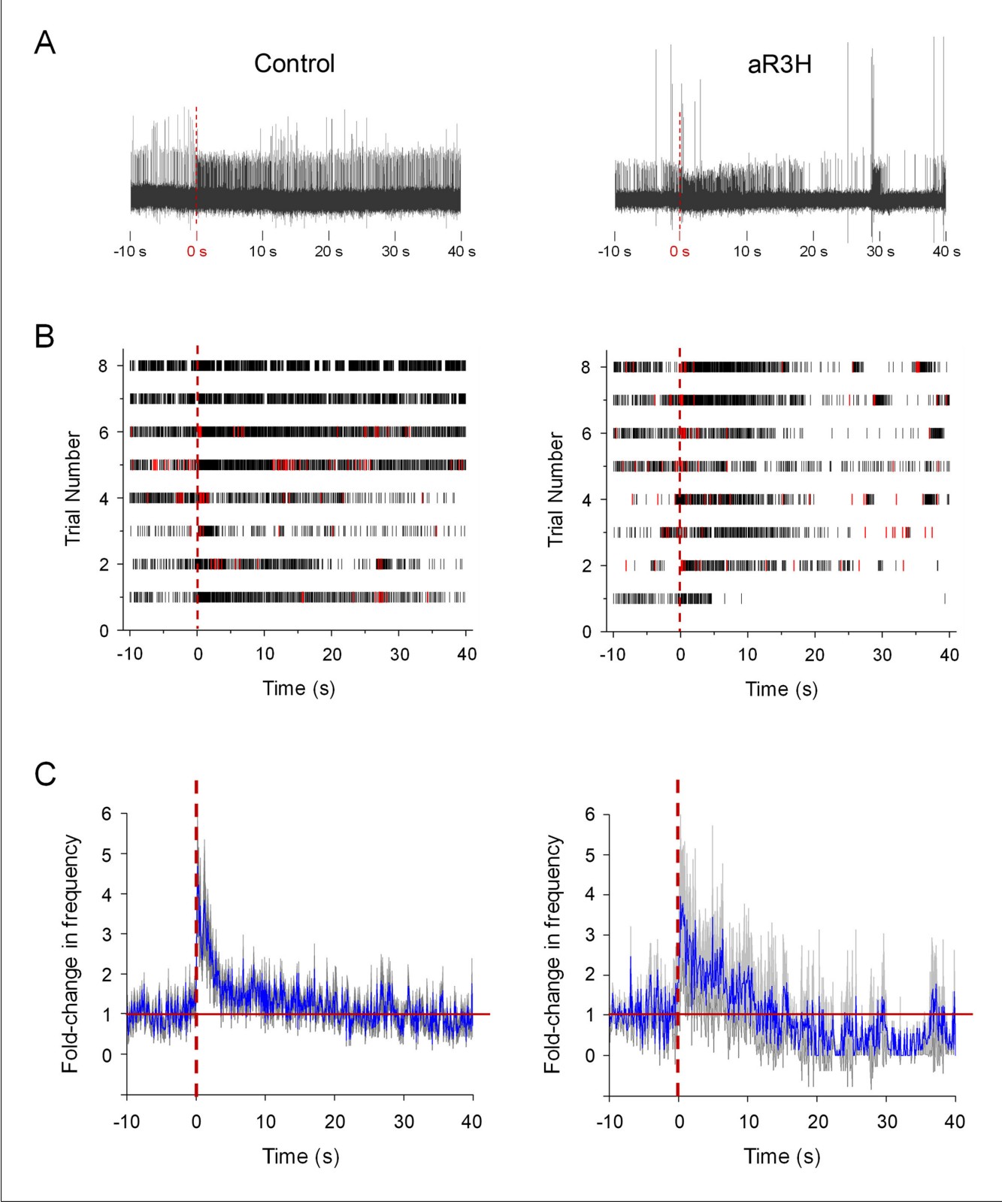

**Figure 3.** aR3H expression results in frequency-dependent hypoexcitability in Purkinje cells. At 5 dpf, live zebrafish were adapted to a LED light, which was turned off at time 0 s (*Hsieh et al., 2014*). (A) Representative recordings from a control Purkinje cell (left) and a Purkinje cell expressing aR3H (right) before and after turning off the LED are shown. Time 0 s is indicated by dashed vertical lines. (B) Raster plots show firing of simple spikes (black) and complex spikes (red) before and after turning off the LED at time 0 s (dashed vertical lines) for control cells (left) and aR3H-expressing cells (right). Each

*Figure 3 continued on next page*

Figure 3 continued

row represents a different trial. Data were obtained from 8 control cells in 5 animals and 8 aR3H-expressing cells in 4 animals. (**C**) The average frequency of tonic firing per 100 ms interval was normalized to the average frequency calculated over the 10 s period before turning the LED off at time 0 s (dashed vertical lines). The fold-change in frequency from all trials was averaged and plotted versus time for control (left) and aR3H-expressing (right) Purkinje cells (blue lines; $n$ = 8 control cells, $n$ = 8 aR3H-expressing cells). The gray shaded areas represent the SEM. The solid red horizontal line indicates the original firing frequency before the imposition of sudden darkness.

caudal primary (CaP) motor neurons in the zebrafish spinal cord (*Issa et al., 2011*). In response to depolarizing current injection, aR3H-expressing motor neurons fail to maintain spiking throughout the pulse, unlike control cells. This effect is more prominent at higher firing frequencies, which result in faster accumulation of $Na^+$ channel inactivation. Decreased excitability is correlated with a reduction in the amplitude of outward currents, consistent with the dominant negative effect of the aR3H mutation (*Issa et al., 2011*; *Waters et al., 2006*; *Minassian et al., 2012*; *Mock et al., 2010*).

Based on these results, we hypothesized that aR3H expression would result in a frequency-dependent decrease in the excitability of cerebellar Purkinje cells. To test this proposal, we increased firing frequency in Purkinje cells in vivo using the visual system. The zebrafish cerebellum receives visual input via mossy fiber pathways by 4 dpf (*Hsieh et al., 2014*). Mossy fibers activate parallel fibers, which in turn synapse on Purkinje cells. We have shown that by 4 dpf, sudden darkness leads to a dramatic increase in tonic firing frequency in the majority of medially-located Purkinje cells (*Hsieh et al., 2014*). In control cells, the firing rate remains elevated for several seconds after the stimulus and then gradually returns to the original tonic firing frequency (*Figure 3A–C*; *Hsieh et al., 2014*). In aR3H-expressing Purkinje cells, sudden darkness elevated the tonic firing frequency by an average of 4-fold, similar to the 4.6-fold increase seen in control cells (*Figure 3C*). However, during the recovery phase of the response, the firing rate in aR3H-expressing cells fell below the original firing frequency, in some cases leading to prolonged cessation of firing (*Figure 3A–C*). In contrast, control cells returned smoothly to the original tonic firing frequency (*Figure 3C*). We conclude that aR3H-expressing Purkinje cells have a latent hypoexcitability that is revealed after an interval of evoked, high frequency firing.

Our results indicate that the aR3H and iR4H mutations have differential effects on Purkinje cell excitability in vivo, with iR4H dramatically and transiently increasing excitability and aR3H decreasing excitability during evoked, high frequency firing, with no significant effect on tonic firing or complex spiking under basal conditions.

## Infant-onset iR4H mutation disrupts Purkinje cell development

In humans, cerebellar development extends into the first postnatal year. In infant-onset SCA13, cerebellar maldevelopment and degeneration are evident by 10 months of age (*Khare et al., 2017*). In contrast, in adult-onset SCA13, progressive locomotor deficits and cerebellar degeneration emerge during the third or fourth decade of life (*Waters et al., 2005*; *Waters et al., 2006*; *Subramony et al., 2013*). To investigate whether infant- and adult-onset SCA13 mutations have differential effects on the morphological maturation and survival of Purkinje cells during cerebellar development in vivo, we expressed aR3H, iR4H, or exogenous wild-type Kv3.3 in Purkinje cells by F0 transgenesis. Expressing cells were identified by the presence of membrane-bound EGFP (mEGFP), which was produced as a separate protein from the same plasmids using a 2A sequence (*Kim et al., 2011*). In control experiments, mEGFP was expressed alone. Using mEGFP to characterize cell morphology was advantageous because the intensity of the membrane-bound reporter protein was well distributed between the cell soma and processes. Individual expressing neurons were repeatedly imaged in live animals using a laser scanning confocal microscope starting on the 3rd day post-fertilization.

The normal course of Purkinje cell development was characterized in mEGFP-expressing control cells. By 3.25 dpf, Purkinje cells had begun to extend processes (*Figure 4A*; *Tanabe et al., 2010*). Twelve hours later, at ~3.75 dpf, the processes were longer and more highly branched. Dendrites could be identified unambiguously by the presence of a high density of spines starting at 3.75 dpf. These results indicate that postsynaptic development is well correlated with functional innervation of Purkinje cells, which receive active parallel fiber inputs from cerebellar granule cells by 4 dpf (*Hsieh et al., 2014*). In addition, functional climbing fiber input from inferior olive neurons begins to

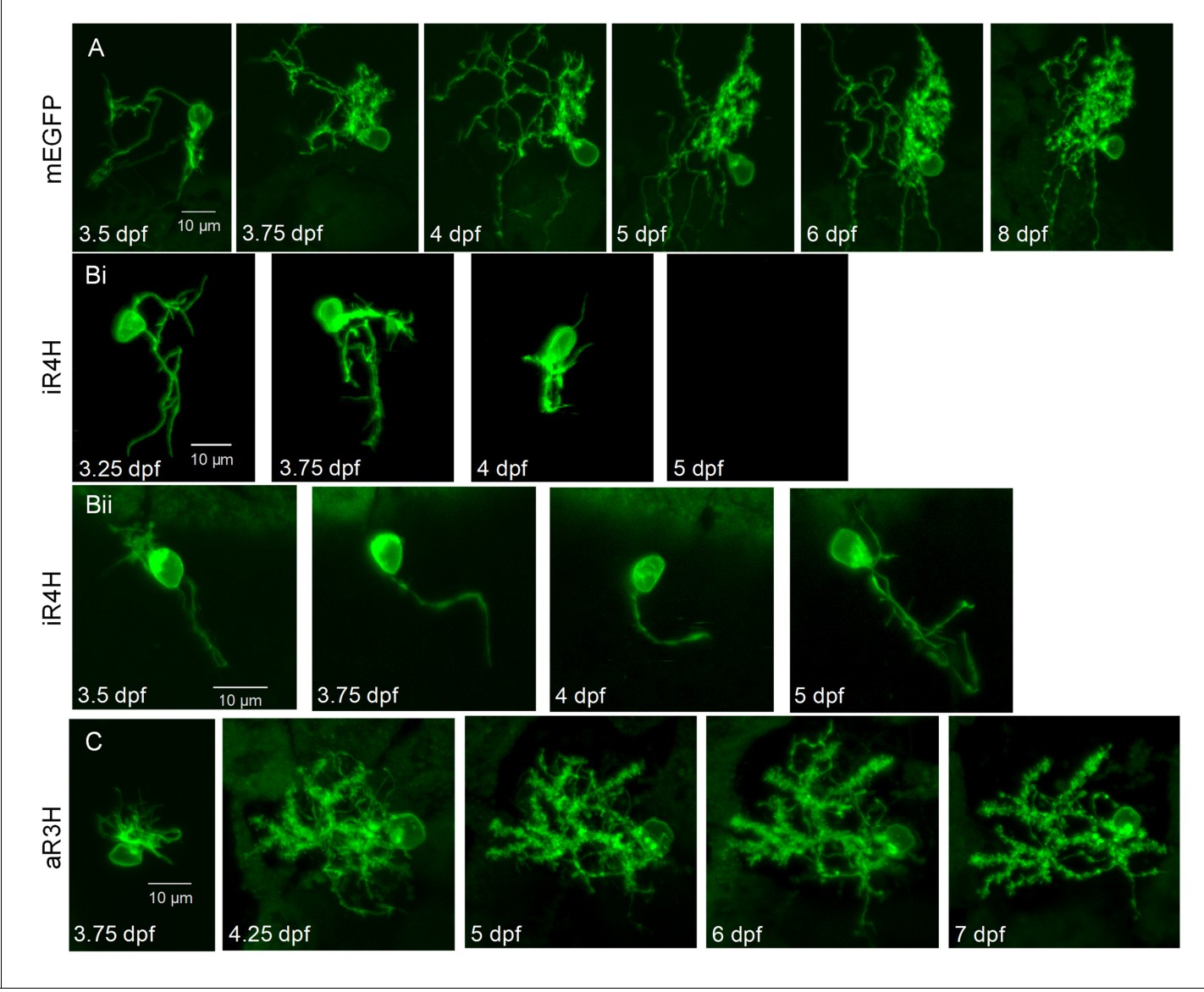

**Figure 4.** Expression of iR4H but not aR3H dramatically disrupts Purkinje cell maturation during cerebellar development in vivo. Live images of individual Purkinje cells during cerebellar development in vivo were acquired using a laser scanning confocal microscope at different times post-fertilization as indicated. Shown are maximum intensity projections of stacks of 1 μm optical sections for A) a control Purkinje cell expressing mEGFP, Bi and Bii) two examples of Purkinje cells expressing iR4H and mEGFP, and C) a Purkinje cell expressing aR3H and mEGFP. In Bi, the cell was no longer visible at 5 dpf. Scale bar: 10 μm.

The online version of this article includes the following figure supplement(s) for figure 4:

**Figure supplement 1.** Expression of exogenous wild-type Kv3.3 (exoWT) does not significantly alter Purkinje cell maturation.

emerge at 4 dpf (*Hsieh et al., 2014*). The complex morphology of Purkinje cells was maintained on subsequent days with ongoing growth and refinement of the dendritic arbor (*Figure 4A*; *Video 1*).

Purkinje cells expressing the infant-onset iR4H mutation began to extend processes on schedule on the 3rd day post-fertilization (*Figure 4B*). However, subsequent growth of the dendritic arbor was limited, and few branches formed (*Figure 4B*; *Video 2*). The density of spines was dramatically decreased compared to control cells, indicating that iR4H severely disrupted postsynaptic development. The relative lack of dendritic spines may contribute to impaired complex spiking in iR4H-expressing Purkinje cells. By 5 dpf, iR4H cells began to disappear (see below). In contrast, the development of aR3H-expressing Purkinje cells was similar to control cells. By 4 dpf, Purkinje cells

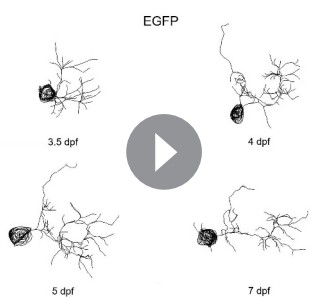

**Video 1.** Purkinje cell expressing mEGFP alone. Live images of a representative Purkinje cell expressing mEGFP were acquired at different times post-fertilization and traced with Imaris software for three-dimensional digital reconstruction. Video shows rotating traces. Dendritic spines are not shown.

https://elifesciences.org/articles/57358#video1

expressing the adult-onset mutation had elaborated complex dendritic arbors studded with spines (*Figure 4C*; *Video 3*). Dendritic morphology was maintained on subsequent days. Purkinje cells expressing exogenous wild-type Kv3.3 also developed a normal morphology (*Figure 4—figure supplement 1*; *Video 4*).

Our results indicate that, in contrast to control cells and cells expressing aR3H or exogenous wild-type Kv3.3, the development of iR4H-expressing neurons was abnormal. To quantitatively compare Purkinje cell development, confocal image stacks were traced with Imaris software for three-dimensional digital reconstruction (*Figure 5*; *Videos 1–4*). When first observed at 3.25 dpf, total process length and the complexity of the dendritic arbor, measured as the number of branches, did not differ significantly between Purkinje cells expressing mEGFP, aR3H, iR4H, or exogenous wild-type Kv3.3 (*Figure 5A,B*). However, iR4H-expressing cells already had the shortest processes and the least complex dendritic arbors. Starting at 3.75 dpf, total process length and the number of branches were reduced significantly in iR4H-expressing cells compared to cells in the other groups. In contrast, process length and branch number in Purkinje cells expressing aR3H, exogenous wild-type Kv3.3, or mEGFP did not differ significantly from each other, although there was a trend for aR3H-expressing cells to have shorter processes and fewer branches by 6.25 dpf. These data indicate that maldevelopment of Purkinje cells was specific to the infant-onset mutation.

## Adult-onset mutation aR3H does not impair postsynaptic development

To investigate whether expression of aR3H or exogenous wild-type Kv3.3 affected postsynaptic development, we estimated spine number per cell as a function of developmental time (*Figure 5C*). iR4H-expressing Purkinje cells were not included in this analysis because there were few, if any, spines on the processes of neurons expressing the infant-onset mutation (*Figure 4B*). The number of spines did not differ significantly in Purkinje cells expressing mEGFP, aR3H, or exogenous wild-type Kv3.3 between 5.25 and 7.25 dpf (*Figure 5C*). There was, however, substantial variation in the numbers of spines in individual cells, which may reflect the dynamic formation and pruning of synapses that occur during cerebellar development. Due to this variation, the sample size was insufficient to

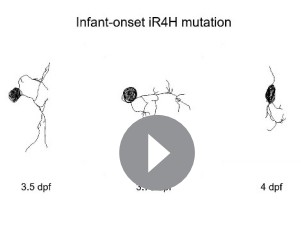

**Video 2.** Purkinje cell expressing iR4H and mEGFP. Live images of a representative Purkinje cell expressing iR4H and mEGFP were acquired at different times post-fertilization and traced with Imaris software for three-dimensional digital reconstruction. Video shows rotating traces. Dendritic spines are not shown.

https://elifesciences.org/articles/57358#video2

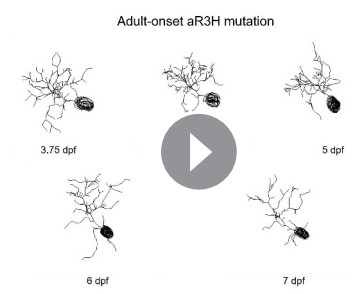

**Video 3.** Purkinje cell expressing aR3H and mEGFP. Live images of a representative Purkinje cell expressing aR3H and mEGFP were acquired at different times post-fertilization and traced with Imaris software for three-dimensional digital reconstruction. Video shows rotating traces. Dendritic spines are not shown.

https://elifesciences.org/articles/57358#video3

detect small changes in the mean number of spines. However, the range of values in cells expressing mEGFP, aR3H, or exogenous wild-type Kv3.3 overlapped (*Figure 5C*). Therefore, we did not find evidence that expression of the adult-onset mutation or exogenous wild-type Kv3.3 impaired postsynaptic development, in contrast to the infant-onset iR4H mutation (*Figure 4B*).

## Infant-onset but not adult-onset mutation disrupts presynaptic development

The scarcity of dendritic spines on iR4H-expressing cells indicates that the infant-onset mutation markedly disrupts postsynaptic development (*Figure 4B*). In parallel, iR4H dramatically reduced branching of the dendritic arbor (*Figure 5B*). Previous work in zebrafish indicates that the formation of stable branches in neuronal

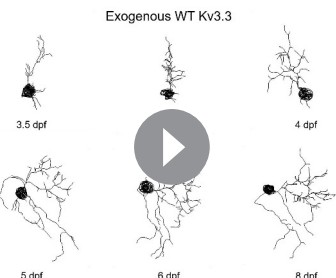

Exogenous WT Kv3.3

3.5 dpf 4 dpf

5 dpf 6 dpf 8 dpf

**Video 4.** Purkinje cell expressing exogenous wild-type Kv3.3 (exoWT) and mEGFP. Live images of representative Purkinje cell expressing exoWT and mEGFP were acquired at different times post-fertilization and traced with Imaris software for three-dimensional digital reconstruction. Video shows rotating traces. Dendritic spines are not shown.
https://elifesciences.org/articles/57358#video4

processes is mechanistically linked to synapse formation in both the pre- and postsynaptic compartments (*Niell et al., 2004*; *Meyer and Smith, 2006*). In developing optic tectum neurons in zebrafish, postsynaptic specializations form on newly extended dendritic filopodia. If the synapse is maintained, the filopodium develops into a stable dendritic branch (*Niell et al., 2004*). Similarly, presynaptic development and branching of the terminal axonal arbor are tightly correlated in retinal ganglion cells in zebrafish (*Meyer and Smith, 2006*). Synapses develop on nascent branches. If a stable synapse is not formed, the branch is retracted.

We investigated whether the SCA13 mutations disrupt synaptic development and branching in the presynaptic compartment using caudal primary (CaP) motor neurons in the zebrafish spinal cord as a model population of cells (*Figure 5—figure supplement 1A*; *Myers et al., 1986*; *Westerfield et al., 1986*). Like Purkinje cells, CaP cells are fast-spiking neurons that endogenously express Kv3.3 (*Issa et al., 2011*). Similarly, Kv3.3 is expressed in motor neurons in mammals (*Brooke et al., 2004*). We expressed EGFP fusion proteins of iR4H or aR3H in CaP neurons under the control of a motor neuron-specific enhancer (*Issa et al., 2012*). In control experiments, EGFP was expressed alone. We quantified presynaptic specializations in the axonal compartment by co-expressing a synaptophysin-mCherry fusion protein that is targeted to synaptic vesicles (*Figure 5—figure supplement 1B*; *Meyer and Smith, 2006*). Images were acquired at 48 hr post-fertilization (hpf) using a laser scanning confocal microscope. Individual neurons were traced using Imaris software for three-dimensional reconstruction and morphological quantification.

Expression of the infant-onset iR4H mutation dramatically reduced the number of synapses in the axonal compartment compared to CaP neurons expressing the adult-onset aR3H mutation or EGFP alone (*Figure 5—figure supplement 1B,C*; *Figure 5—figure supplement 2*). In contrast, the number of synapses in CaP neurons expressing aR3H did not differ significantly from EGFP controls. Furthermore, the axons of iR4H-expressing motor neurons had significantly fewer distal branches than the other groups (*Figure 5—figure supplement 1B,D*; *Figure 5—figure supplement 2*), reminiscent of reduced dendritic branching in iR4H-expressing Purkinje cells. The length of the distal branches that were present in iR4H-expressing CaP neurons did not differ significantly from those in EGFP-expressing motor neurons, although they were significantly shorter than those in aR3H-expressing cells (*Figure 5—figure supplement 1E*). In contrast, the length of distal branches in aR3H-expressing neurons and EGFP controls did not differ significantly. The paucity of distal branches in iR4H-expressing CaP neurons was not due to a failure of axonal outgrowth because the length of the main axon shaft did not differ significantly in motor neurons expressing iR4H, aR3H, or EGFP alone (*Figure 5—figure supplement 1B,F*; *Figure 5—figure supplement 2*).

Given the relationship between synaptogenesis and branching in zebrafish neurons (*Niell et al., 2004*; *Meyer and Smith, 2006*), our results suggest that reductions in presynaptic specializations and distal branching in iR4H-expressing CaP neurons are not independent phenomena. Rather, both

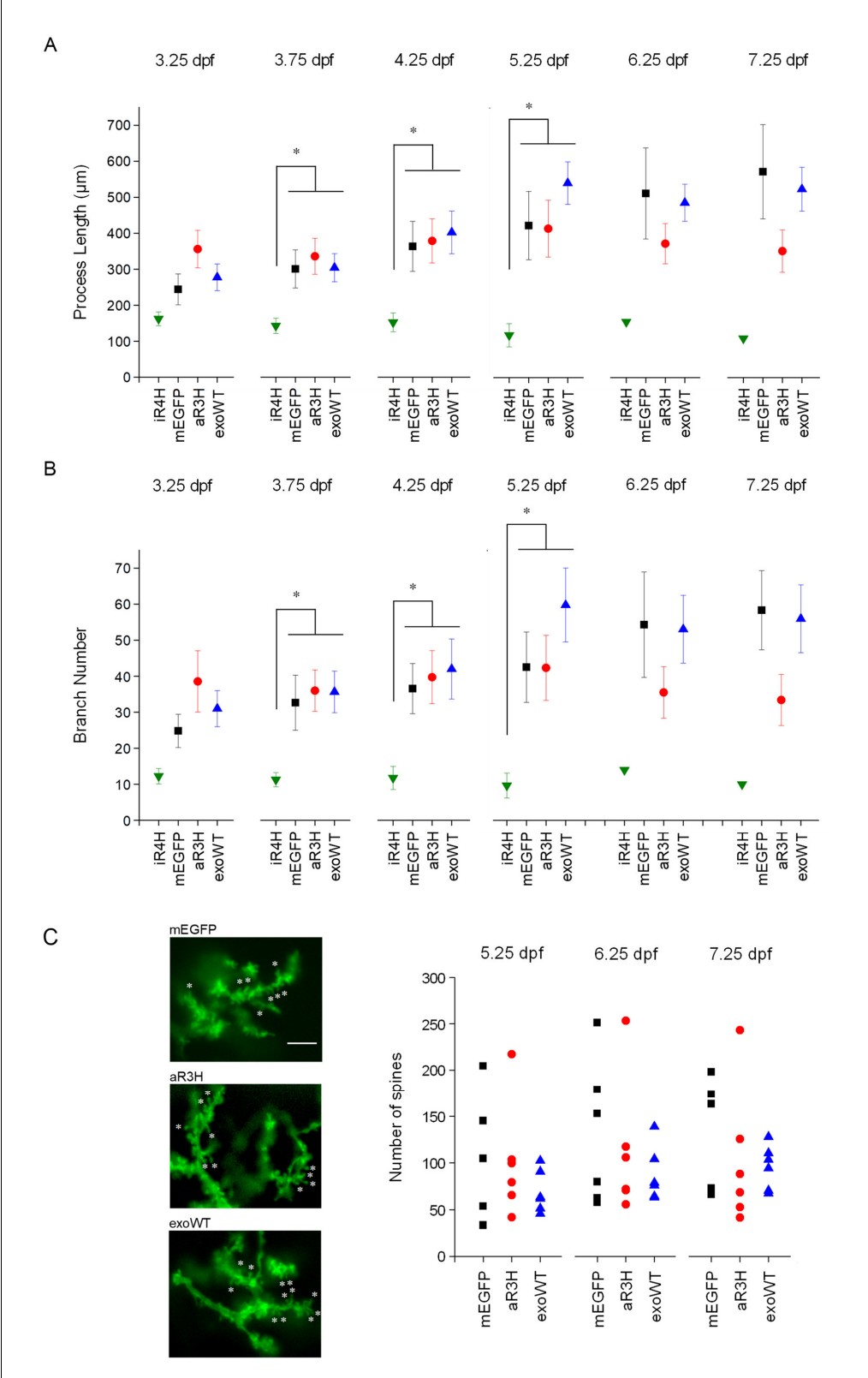

**Figure 5.** iR4H but not aR3H or exoWT significantly alters Purkinje cell maturation and postsynaptic development during cerebellar development in vivo. Images of individual Purkinje cells expressing mEGFP alone (mEGFP) (*n* = 11) or in combination with iR4H (*n* = 12), aR3H (*n* = 11), or exogenous wild type Kv3.3 (exoWT) (*n* = 11) were traced and quantified using Imaris software. (**A**) Total process length at different times post-fertilization is shown. Data are provided as mean ± SEM. Note that there were too few iR4H-expressing cells to calculate SEM values after 5.25 dpf (see **Figure 6A**). (**B**) The
*Figure 5 continued on next page*

*Figure 5 continued*

total number of branches at different times post-fertilization is shown. Data are provided as mean ± SEM. Note that there were too few iR4H-expressing cells to calculate SEM values after 5.25 dpf (see *Figure 6A*). In A and B, statistical analyses were performed using SPSS software (IBM, Armonk NY). Linear mixed model was applied to $log_{10}$-transformed data followed by Bonferroni post-hoc test. *, p<0.05. At 3.75, 4.25, and 5.25 dpf, values for iR4H-expressing cells differed significantly from those for mEGFP-, aR3H-, and exoWT-expressing cells. Values for mEGFP-, aR3H-, and exoWT-expressing cells did not differ significantly from each other. Statistical analysis for iR4H was not performed at 6.25 and 7.25 dpf due to the small number of surviving cells. (C) Left: Images acquired at 7.25 dpf show dendritic spines in Purkinje cells expressing mEGFP alone (top) or with aR3H (middle) or exoWT (bottom). A subset of spines is marked with '*'. Scale bar = 5 μm. Right: Dendritic spines were counted in triplicate using ImageJ at 5.25, 6.25, and 7.25 dpf for cells expressing mEGFP alone (*n* = 6) or with aR3H (*n* = 6) or exoWT (*n* = 6). The three values were averaged to estimate the number of spines in individual cells. Each symbol indicates the averaged spine count for a single cell. Mean spine numbers from all the cells in each group ± SEM were: mEGFP-expressing cells, 96 ± 28 at 5.25 dpf, 130 ± 32 at 6.25 dpf, 124 ± 25 at 7.25 dpf; aR3H-expressing cells, 101 ± 25 at 5.25 dpf, 112 ± 30 at 6.25 dpf, 103 ± 30 at 7.25 dpf; exoWT-expressing cells, 69 ± 9 at 5.25 dpf, 87 ± 12 at 6.25 dpf, 96 ± 10 at 7.25 dpf. The mean number of spines did not differ significantly at any time point in cells expressing mEGFP alone or with aR3H or exoWT. iR4H-expressing cells were not included in this analysis because of the scarcity of dendritic spines (see *Figure 4B*).

The online version of this article includes the following figure supplement(s) for figure 5:

**Figure supplement 1.** iR4H but not aR3H significantly impairs presynaptic development.

**Figure supplement 2.** iR4H disrupts distal branching and synaptogenesis in the axonal compartment of CaP motor neurons.

effects likely reflect significantly impaired synaptogenesis. Taken together, our results in Purkinje cells and CaP motor neurons indicate that the iR4H mutation disrupts both pre- and post-synaptic development, impairing the formation of stable branches in the axonal and dendritic arbors, respectively.

Proximal axonal branches, located at or dorsal to the midline, are present in some CaP neurons (*Figure 5—figure supplement 1B,G*). Although the number of these branches was similar in all three groups, the proximal branches in iR4H-expressing cells were significantly longer than those seen in CaP neurons expressing aR3H or EGFP (*Figure 5—figure supplement 1G,H*; *Figure 5—figure supplement 2* ). Interestingly, the effects of the iR4H mutation on proximal and distal branching in CaP axons were similar to those of another infant-onset mutation, F448L (iFL) (*Figure 1A*; *Issa et al., 2012*). iFL-expressing CaP neurons also have fewer distal branches than control cells and extend long, aberrant proximal collaterals (*Issa et al., 2012*). The iFL mutation causes dominant changes in Kv3.3 gating similar to those of iR4H, but iFL does not have an accompanying dominant negative effect (*Minassian et al., 2012*). Therefore, it is likely that altered gating and resulting changes in excitability underlie the similar effects of these two infant-onset mutations on CaP development.

## Infant-onset but not adult-onset mutation causes Purkinje cell loss during cerebellar development

We observed that iR4H-expressing Purkinje cells disappeared during cerebellar development in vivo (*Figure 6A*). This striking phenotype was highly penetrant. By 5 dpf, one third of the iR4H-expressing Purkinje cells that had been imaged at 4 dpf were no longer visible, and by 6 dpf, more than 80% of the cells had vanished. Two cells persisted until 7 dpf, one of which was still visible at 8 dpf prior to disappearing. To verify that the loss of iR4H-expressing Purkinje cells corresponded to cell death, acridine orange was added to the aquarium. Acridine orange is a fluorescent molecule that is able to cross the plasma membranes of living and early apoptotic cells. It intercalates into DNA, staining the nucleus green. The presence of bright green puncta indicates the formation of condensed and fragmented chromatin, a hallmark of apoptotic cells (*Atale et al., 2014*). At 5 dpf, uptake of acridine orange into iR4H-expressing cells and labeling of puncta of condensed, fragmented chromatin were evident, signifying the activation of cell death pathways (*Figure 6—figure supplement 1*; *Atale et al., 2014*). In contrast to iR4H-expressing cells, all Purkinje cells expressing aR3H, exogenous wild-type Kv3.3, or mEGFP alone that were imaged at 3.75 dpf remained visible through 8 dpf, the latest time point examined, with no evidence of net process retraction or atrophy (*Figure 6A*, see *Figure 4A,C*, and *Figure 4—figure supplement 1*). These results indicate that expression of the infant-onset mutation, but not the adult-onset mutation or exogenous wild-type Kv3.3, resulted in rapid Purkinje cell death during cerebellar development in vivo. The age-dependent degeneration of iR4H-expressing Purkinje cells in zebrafish is reminiscent of the early postnatal degeneration of the cerebellum that occurs in infant-onset SCA13 patients (*Khare et al., 2017*).

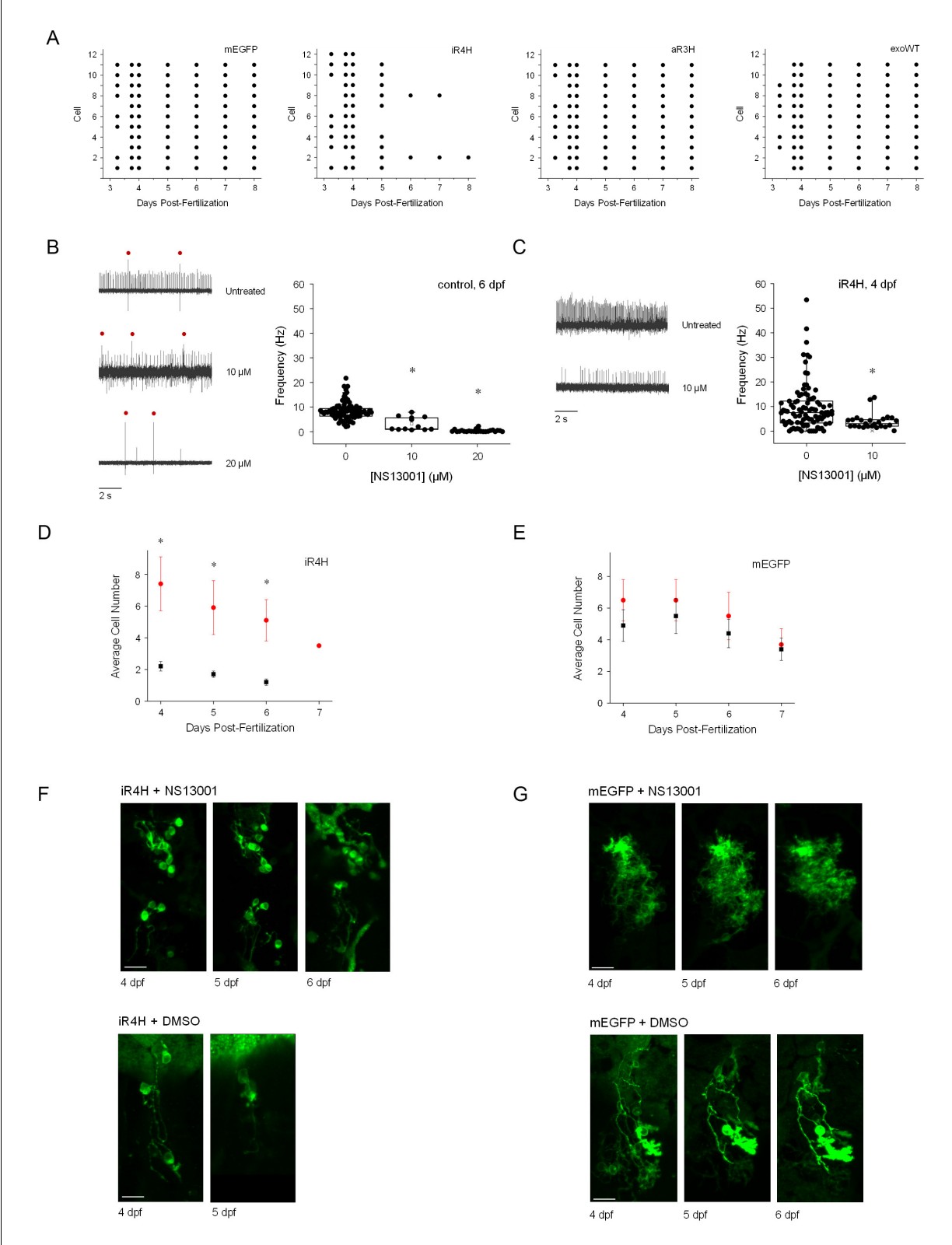

**Figure 6.** Suppression of hyperexcitability increases the early survival of iR4H-expressing Purkinje cells. (A) Individual Purkinje cells expressing (from left to right) mEGFP alone or with iR4H, aR3H, or exogenous wild-type Kv3.3 (exoWT) were repeatedly imaged in living zebrafish at different times post-fertilization as indicated. Each row corresponds to an individual cell. Each symbol indicates the persistence of individual cells at different observation times during cerebellar development. (B) Acute application of 10 or 20 μM NS13001 during patch clamp recording decreased firing frequency in control

*Figure 6 continued on next page*

*Figure 6 continued*

Purkinje cells at 6 dpf. Left, representative traces recorded in the absence (top) or presence of 10 µM (middle) or 20 µM (bottom) NS13001 are shown. Red dots: complex spikes. Right, average frequencies of tonic firing in control cells in 0, 10, and 20 µM NS13001 are shown. Each symbol represents a trace. Mean firing frequencies ± SEM were 8.5 ± 0.4 ($n$ = 90), 3.2 ± 0.8 ($n$ = 12), and 0.3 ± 0.1 ($n$ = 24) in 0, 10, or 20 µM drug, respectively. Firing rates in 0, 10, and 20 µM NS13001 differed significantly by one-way ANOVA (F = 69.68, p=$5.57×10^{-21}$) with post-hoc analysis by t-test ($p$ values: 0 µM vs. 10 µM, $3.64 × 10^{-6}$; 0 µM vs. 20 µM, $1.04 × 10^{-19}$; 10 µM vs. 20 µM, $1.42 × 10^{-5}$). (C) Acute application of 10 µM NS13001 during patch clamp recording decreased firing frequency in iR4H-expressing Purkinje cells at 4 dpf. Left, representative traces recorded in the absence (top) or presence of 10 µM (bottom) NS13001 are shown. Right, average frequencies of tonic firing in iR4H-expressing cells in 0 and 10 µM NS13001 are shown. Each symbol represents a trace. Mean firing frequencies ± SEM were 9.7 ± 1 ($n$ = 92) and 3.9 ± 0.6 ($n$ = 25) in 0 and 10 µM NS13001, respectively. Firing rates in 0 and 10 µM NS13001 differed significantly by two-sided unpaired t-test, p=by $3.78 × 10^{-3}$. Firing frequencies in control and iR4H-expressing cells are not directly comparable because they were measured at different times post-fertilization. (D) and (E) Zebrafish were injected with plasmids encoding D) iR4H and mEGFP or E) mEGFP alone. Animals were chronically treated with 20 µM NS13001 (red symbols) or an equivalent amount of DMSO, the drug vehicle (black symbols), added to the aquarium water starting at 3.25 dpf. The number of Purkinje cells expressing (D) iR4H and mEGFP or (E) mEGFP alone was counted in each animal at different times post-fertilization as indicated. Data are presented as the mean number of expressing cells per animal ± SEM. Treatment with NS13001 significantly increased the number of iR4H-expressing Purkinje cells compared to DMSO treatment alone. *, p<0.05 by SPSS linear mixed model with Bonferroni post-hoc test. (F) Representative images of iR4H-expressing Purkinje cells in animals chronically treated with NS13001 (upper) or DMSO alone (lower) at different times post-fertilization, as indicated. (G) Representative images of mEGFP-expressing Purkinje cells in animals chronically treated with NS13001 (upper) or DMSO alone (lower) at different times post-fertilization, as indicated.

The online version of this article includes the following figure supplement(s) for figure 6:

**Figure supplement 1.** iR4H-expressing Purkinje cells die during cerebellar development in vivo.

## Suppressing hyperexcitability increases the early survival of iR4H-expressing Purkinje cells

Our data indicate that transient hyperexcitability in iR4H-expressing Purkinje cells is temporally correlated with highly penetrant maldevelopment followed by cell death. To investigate the role of hyperexcitability in the degeneration of iR4H-expressing neurons, we used NS13001, a semi-selective agonist of small conductance, $Ca^{2+}$-activated $K^+$ (SK) channels (*Kasumu et al., 2012*; *Coleman et al., 2014*). Increased current through these channels is expected to reduce neuronal excitability (*Kasumu et al., 2012*). In a mouse model of SCA2, oral treatment with NS13001 regularizes Purkinje cell firing, improves locomotor function, and reduces Purkinje cell loss (*Kasumu et al., 2012*).

We first investigated whether activation of SK channels by NS13001 suppressed tonic firing in zebrafish Purkinje cells. When acutely applied to the exposed zebrafish cerebellum during patch clamp recordings at 6 dpf, 10 or 20 µM NS13001 significantly reduced the frequency of spontaneous tonic firing in Purkinje cells in control animals (*Figure 6B*). Application of 10 µM NS13001 reduced spike frequency by approximately 60%, and 20 µM nearly abolished firing. We then assessed whether acute treatment of the exposed cerebellum with the drug suppressed hyperexcitability in Purkinje cells expressing the iR4H mutant subunit. Application of 10 µM NS13001 at 4 dpf reduced the mean firing rate by approximately 60%, similar to control cells (*Figure 6C*). Note that the firing rates in control and iR4H-expressing cells are not directly comparable because the recordings were obtained at different times post-fertilization.

To determine whether chronic treatment with the drug improved the survival of iR4H-expressing Purkinje cells during cerebellar development in vivo, 20 µM NS13001 or an equivalent amount of the drug vehicle, DMSO, was added to the zebrafish aquarium starting at 3.25 dpf. Drug-treated zebrafish were noticeably more sluggish than fish treated with vehicle alone, indicating that NS13001 was able to penetrate the skin. At intervals, live images of Purkinje cells were obtained using a laser scanning confocal microscope. We observed that chronic drug treatment significantly increased the number of iR4H-expressing neurons compared to DMSO alone, so that the average number of mEGFP- and iR4H-expressing Purkinje cells in drug-treated animals did not differ significantly between 4 and 7 dpf (*Figure 6D,E*). In contrast, after treatment with DMSO alone, zebrafish contained a significantly higher number of mEGFP-expressing than iR4H-expressing Purkinje cells (*Figure 6D,E*).

Importantly, NS13001 treatment did not significantly increase the number of mEGFP-expressing Purkinje cells compared to DMSO alone (*Figure 6E*). This is consistent with the conclusion that the increased number of iR4H-expressing neurons observed during drug treatment reflected enhanced survival, rather than increased proliferation. Therefore, our data indicate that, in the absence of

NS13001, a significant number of iR4H-expressing Purkinje cells died rapidly, before they could be observed under the microscope. These results suggest that hyperexcitability contributes to the early demise of Purkinje cells expressing the infant-onset mutation during cerebellar development in vivo.

It was not feasible to determine how long iR4H-expressing Purkinje cells could survive after reducing excitability. Between 4 and 7 dpf, treatment with NS13001 or DMSO alone resulted in a loss of Purkinje cells expressing either mEGFP or iR4H (*Figure 6D,E*). This suggests that vehicle and/or drug treatment reduced Purkinje cell viability independent of the iR4H mutation. Due to the apparent toxicity of chronic exposure to NS13001 and/or DMSO, some treated animals did not survive until 7 dpf.

Despite prolonged early survival of neurons expressing the infant-onset mutation, chronic treatment with NS13001 did not restore normal morphological maturation in iR4H-expressing Purkinje cells, as evidenced by the lack of well-developed dendritic arbors (*Figure 6F*). In contrast, mEGFP-expressing neurons extended processes in the presence of NS13001 (*Figure 6G*).

## Discussion

### Infant- and adult-onset SCA13 mutations have opposite effects on excitability and differentially affect Purkinje cell maturation and survival during cerebellar development in vivo

To understand pathogenic mechanisms that underlie SCA13, it is essential to identify distinct effects of infant- and adult-onset mutations in cerebellar neurons that could account for the existence of two clinical forms of the disease (*Herman-Bert et al., 2000*; *Waters et al., 2005*; *Waters et al., 2006*; *Figueroa et al., 2010*; *Figueroa et al., 2011*; *Duarri et al., 2015*). Because Kv3.3 regulates spontaneous pacemaking and evoked high frequency firing in Purkinje cells (*Khaliq et al., 2003*; *Martina et al., 2003*; *Martina et al., 2007*; *Akemann and Knöpfel, 2006*), we tested the hypothesis that early- and late-onset SCA13 mutations alter Purkinje cell excitability in distinct ways. In parallel, we investigated whether the mutations differentially affect the maturation and viability of Purkinje cells during cerebellar development in vivo in zebrafish.

We found that the infant-onset iR4H mutation resulted in transient hyperexcitability that emerged soon after Purkinje cells became spontaneously active. Transient hyperexcitability was temporally correlated with maldevelopment, including stunted process extension, dramatically decreased complexity of the dendritic arbor, and impaired synaptogenesis, followed by highly penetrant, rapid cell death during cerebellar development. Developmental abnormalities and death occurred in well-isolated iR4H-expressing neurons, indicating that cell autonomous mechanisms were responsible. The effects of iR4H on Purkinje cells during development are reminiscent of the maldevelopment and atrophy of the cerebellum during infancy in early-onset SCA13, suggesting that pathogenesis in humans and zebrafish are mechanistically related (*Waters et al., 2006*; *Khare et al., 2017*).

By comparing the functional development of the cerebellum, the period of vulnerability of iR4H-expressing Purkinje cells in zebrafish can be specifically correlated with the early postnatal period in mammals, corresponding to the second and third postnatal weeks in rodents (*Fry, 2006*; *Gruol et al., 2005*; *Mariani and Changeux, 1981*). In zebrafish, activity in the cerebellar circuit develops rapidly (*Hsieh et al., 2014*). Spontaneous firing in Purkinje cells emerges at 3 to 4 dpf followed by an increase in firing frequency and regularity by 5 dpf (*Hsieh et al., 2014*). By 4 dpf, Purkinje cells receive functional input from parallel fibers, and active climbing fiber input begins to be evident. By 5 dpf, climbing fiber innervation is at a maximum (*Hsieh et al., 2014*). Over the next ~48 hr, weak climbing fiber inputs are winnowed. In mice, the cerebellar circuit becomes functional between postnatal day 5 (P5) and P15. Spontaneous Purkinje cell firing begins to emerge at ~P5 (*Fry, 2006*). Spike frequency increases until ~P12 (*Fry, 2006*). In rats, Purkinje cells are innervated by multiple climbing fibers between P4 and P6 (*Gruol et al., 2005*; *Mariani and Changeux, 1981*). Climbing fibers are then winnowed to a single strong input by P15 (*Mariani and Changeux, 1981*). Thus, iR4H affects the development and survival of zebrafish Purkinje cells during a time that is equivalent to the early postnatal period in mammals. In humans, cerebellar development continues through the first year of life. Our results are therefore consistent with the timing of cerebellar maldevelopment and atrophy during the first postnatal year in patients with infant-onset SCA13 (*Khare et al., 2017*).

In contrast, Purkinje cells expressing the adult-onset aR3H mutation matured normally and survived robustly during cerebellar development in vivo. These results are consistent with the clinical phenotype of late-onset SCA13, in which progressive locomotor deficits and progressive cerebellar degeneration are not evident until adulthood (*Subramony et al., 2013*; *Waters et al., 2005*; *Waters et al., 2006*). Consistent with its relative benignity during cerebellar development, the adult-onset aR3H mutation did not alter the basal excitability of Purkinje cells. The effects of aR3H on firing became apparent only after an interval of evoked, high frequency spiking, when a latent reduction in excitability was revealed. A similar, frequency-dependent hypoexcitability has been reported in several types of neurons after pharmacological or genetic reduction of Kv3 channel activity (*Rudy and McBain, 2001*; *Issa et al., 2011*; *Erisir et al., 1999*; *Lau et al., 2000*; *Macica et al., 2003*).

Expression of exogenous wild-type Kv3.3 did not significantly alter Purkinje cell maturation, excitability, or viability, indicating that the SCA13 mutations are responsible for the phenotypes that we have observed.

Taken together, our results indicate that increasing or decreasing Kv3.3 activity by expressing exogenous wild type or dominant negative aR3H subunits, respectively, does not alter basal excitability in Purkinje cells. Mammalian Purkinje cells express Kv3.3 at high levels; only a fraction of the channels opens during an action potential (*Martina et al., 2003*; *Martina et al., 2007*). During basal tonic firing, a similar spare capacity in zebrafish Purkinje cells could suffice to buffer the spike frequency against the changes in Kv3 current amplitude that occur in our experiments. We did not determine whether action potential amplitude or duration were altered by the expression of SCA13 mutant subunits or exogenous wild type Kv3.3 because the loose patch recordings were not suitable for detailed analysis of spike waveform.

In contrast to our results, Namikawa et al. recently reported that Purkinje cells expressing the adult-onset aR3H mutation died starting between 4 and 6 dpf in zebrafish (*Namikawa et al., 2019*). Cell death during this early period of cerebellar development is incompatible with the delayed onset of SCA13 in human patients carrying the aR3H mutation (*Subramony et al., 2013*; *Waters et al., 2005*; *Waters et al., 2006*). One possibility is that the abnormally early death of aR3H-expressing cells in that study resulted from overexpression of the mutant protein (*Namikawa et al., 2019*). Expression was driven by multiple copies of a Purkinje cell-specific enhancer sequence. Strikingly, the onset of death in aR3H-expressing cells depended on expression level, as controlled by the number of copies of the enhancer element. Increasing expression of aR3H significantly accelerated the onset of Purkinje cell death (*Namikawa et al., 2019*). In contrast, we used the native promoter of the *aldoca* gene, which is specifically expressed in zebrafish Purkinje cells starting early in cerebellar development. As discussed below, the differential effects of aR3H and iR4H on Purkinje cell excitability strongly suggest that significant overexpression was not a confounding factor in our experiments. This likely accounts for the differences between our results and those of Namikawa et al., although we cannot rule out other possibilities, such as expression in different subsets of Purkinje cells due to the use of different drivers (*Namikawa et al., 2019*).

## Changes in Purkinje cell excitability are consistent with effects of infant- and adult-onset mutations on Kv3.3 function

The aR3H and iR4H mutations are separated by just two amino acids in the S4 transmembrane segment in the Kv3.3 voltage sensor domain (*Figure 1A*). Both mutations replace an arginine residue with a histidine. Despite these similarities, aR3H and iR4H are associated with distinct SCA13 clinical phenotypes and have opposite effects on Purkinje cell excitability (*Waters et al., 2006*; *Figueroa et al., 2010*; *Figueroa et al., 2011*).

The aR3H and iR4H subunits are non-functional when expressed alone (*Waters et al., 2006*; *Minassian et al., 2012*). To alter Purkinje cell excitability, mutant subunits must assemble with endogenous wild-type subunits to form tetrameric channels. We have reported that one aR3H or one iR4H subunit can assemble with three wild-type subunits to form functional, cell surface channels, but additional subunits result in dominant negative suppression of channel activity (*Minassian et al., 2012*). This suppression is likely mediated by intracellular retention of channel complexes that contain multiple mutant subunits (*Schulteis et al., 1998*; *Khare et al., 2017*; *Gallego-Iradi et al., 2014*). The crucial difference between the aR3H and iR4H mutations is evident when one mutant subunit assembles with three wild-type subunits. The aR3H mutation does not

affect the steady state or kinetic properties of Kv3.3 channels when measured under physiologic ionic conditions (*Minassian et al., 2012*). Therefore, aR3H acts as a simple dominant negative subunit. Patients with this mutation are expected to have a reduced amplitude of Kv3 current with normal functional properties. In contrast, incorporation of an iR4H subunit results in dominant gain-of-function alterations in channel gating (*Minassian et al., 2012*). The mutant subunit shifts the voltage dependence of channel opening modestly but significantly in the hyperpolarized direction. In addition, there are small changes in activation and deactivation kinetics when measured under physiologic ionic conditions. Incorporation of multiple iR4H subunits in the tetramer, however, results in dominant negative suppression of current amplitude (*Minassian et al., 2012*). The ratio of iR4H to wild-type subunits in the tetramer therefore determines whether the mutation exerts dominant gain- or loss-of-function effects on Kv3.3. Patients with the iR4H mutation would be expected to have a reduced amplitude of Kv3 current but the gating properties of the residual current would be abnormal.

The opposite changes in excitability that result from expressing aR3H or iR4H in Purkinje cells are consistent with the distinct effects of these mutations on Kv3.3 function. Reducing Kv3.3 activity, whether by dominant negative suppression, channel blockade, or genetic deletion, results in frequency-dependent hypoexcitability (*Rudy and McBain, 2001*; *Issa et al., 2011*; *Erisir et al., 1999*; *Lau et al., 2000*). Although it may seem paradoxical that decreasing $K^+$ channel activity reduces excitability, this reflects the specialized gating properties of Kv3 channels. Kv3 channels activate in a depolarized voltage range (*Rudy and McBain, 2001*). They are not involved in maintaining the resting potential or suppressing excitability near threshold. Instead, Kv3 channels open with fast kinetics during action potentials. As a result, the membrane rapidly repolarizes, which effectively removes $Na^+$ channel inactivation (*Rudy and McBain, 2001*). Reducing Kv3 activity increases the accumulation of $Na^+$ channel inactivation during repetitive firing (*Rudy and McBain, 2001*). Decreases in excitability are frequency dependent because $Na^+$ channel inactivation accumulates more rapidly at higher firing rates.

In contrast, iR4H increases Purkinje cell excitability. When present in active, cell surface channels, the iR4H subunit shifts the voltage dependence of activation in the hyperpolarized direction (*Minassian et al., 2012*). Although again it may seem paradoxical that shifting the voltage dependence of opening in the negative direction increases excitability, it is important to emphasize the relatively small amplitude of the shift. Channels containing iR4H subunits still have a low open probability at the resting potential and at threshold (*Minassian et al., 2012*). As a result, action potential initiation in Purkinje cells would not be significantly impaired. Rather, due to the shift in gating, iR4H-containing channels would open earlier during an action potential leading to more rapid repolarization and briefer spikes. As a result, open channel block and inactivation of $Na^+$ channels would be reduced, increasing $Na^+$ channel availability (*Carter and Bean, 2009*). Due to faster repolarization, open channel block of $Na^+$ channels would be relieved at an earlier time. The resulting resurgent $Na^+$ current and the greater availability of $Na^+$ channels would then trigger the next action potential sooner than normal, increasing the firing frequency. The lack of a complex dendritic arbor in iR4H-expressing Purkinje cells would decrease neuronal capacitance, which may exacerbate increases in excitability caused by altered Kv3.3 gating. Additionally, compensatory changes in the expression of other channels may modulate the firing properties that we have observed (*Khaliq et al., 2003*).

The observation that iR4H dramatically increases the excitability of Purkinje cells provides strong evidence that significant overexpression is not a confounding factor in our experiments. If iR4H subunits were present in large excess over endogenous wild-type subunits, most channel tetramers would contain multiple mutant subunits. At that stoichiometry, iR4H would cause dominant negative suppression of channel activity, which would result in frequency-dependent hypoexcitability. With overexpression, active tetramers containing only one mutant subunit would be extremely rare, dramatically reducing the number of cell surface channels with altered gating that mediate hyperexcitability.

Importantly, several other infant-onset mutations cause similar changes in Kv3.3 gating, shifting the voltage dependence of opening modestly in the hyperpolarized direction (*Waters et al., 2006*; *Minassian et al., 2012*; *Duarri et al., 2015*). In contrast to iR4H, these mutant subunits form active channels in the absence of wild-type Kv3.3 and do not mediate dominant negative suppression of current amplitude (*Waters et al., 2006*; *Minassian et al., 2012*; *Duarri et al., 2015*). These data

suggest that altered gating is an essential aspect underlying infant-onset SCA13. This is consistent with the idea that abnormally increased excitability may be an important factor in the etiology of this form of the disease.

## Different pathogenic mechanisms likely contribute to infant- and adult-onset SCA13

Our results suggest that different pathogenic mechanisms contribute to cerebellar degeneration in infant- and adult-onset SCA13. We propose that hyperexcitability contributes to maldevelopment and degeneration of the cerebellum in infant-onset SCA13. In our experiments, iR4H-expressing cells became hyperexcitable prior to cell death. Large increases in spike frequency by 4.25 dpf occurred concurrently with significant disruption of Purkinje cell development. Abnormal firing peaked during the 4th day post-fertilization, whereas neuronal loss occurred primarily at 5 to 6 dpf, suggesting that increased firing triggered the activation of cell death pathways. Before disappearing, Purkinje cells expressing the infant-onset iR4H mutation stopped firing, which might reflect metabolic failure, increased membrane permeability, and/or the onset of cell death. This late stage decline in excitability did not prevent Purkinje cell loss, suggesting that irreversible activation of cell death pathways had already occurred. Consistent with the proposal that hyperexcitability is mechanistically linked to Purkinje cell loss, suppressing firing with the SK channel activator NS13001 increased the early survival of iR4H-expressing neurons. In contrast, treatment with NS13001 did not restore normal morphological maturation, suggesting that different pathways contribute to aberrant development and cell death.

Because it regulates multiple signaling pathways, $Ca^{2+}$ is a plausible intermediate between hyperexcitability and the maldevelopment and death of iR4H-expressing Purkinje cells. High frequency firing is expected to increase $Ca^{2+}$ entry, which may in turn induce $Ca^{2+}$-activated $Ca^{2+}$ release from internal stores. $Ca^{2+}$-activated signaling pathways are essential for normal Purkinje cell development in the early postnatal period (*Gruol et al., 2005*). Altered $Ca^{2+}$ handling has been implicated in neuronal cell death in a variety of neurodegenerative diseases (*Glaser et al., 2019*; *Bernard-Marissal et al., 2018*; *Patron et al., 2018*). Excessive cytoplasmic $Ca^{2+}$ may be transported into mitochondria, triggering irreversible opening of the mitochondrial transition pore and activating cell death pathways (*Rao et al., 2014*; *Bano and Ankarcrona, 2018*; *Abeti and Abramov, 2015*).

Several caveats apply to the NS13001 experiments. It was not possible to determine how effectively external application of NS13001 suppressed hyperexcitability in the cerebellum in vivo, what drug concentration was achieved in the vicinity of Purkinje cells, or how long it took to reach a steady state drug concentration. As a result, the drug may not have been completely effective at decreasing excitability in our experiments. Using higher concentrations of NS13001 was not feasible due to its toxicity and because zebrafish had difficulty swimming and maintaining an upright posture during drug treatment. Our results leave open the possibility that hyperexcitability is not the only mechanism that contributes to the demise of iR4H-expressing Purkinje cells.

In contrast, frequency-dependent hypoexcitability caused by the aR3H mutation did not disrupt Purkinje cell maturation or survival during cerebellar development. Our results do not address how this mutation leads to cerebellar degeneration during adulthood in SCA13 patients. However, the available evidence indicates that frequency-dependent hypoexcitability may be insufficient to trigger late-onset degeneration of the cerebellum. In mice, combined homozygous knockout of *Kcnc3* and *Kcnc1*, the genes that encode Kv3.3 and Kv3.1, respectively, is functionally reminiscent of the aR3H dominant negative mutation, which suppresses the activity of wild-type Kv3.3 and any other co-expressed Kv3 subunits. Although the double knockout exhibits frequency-dependent hypoexcitability (*Matsukawa et al., 2003*), cerebellar architecture is normal with no evidence of degeneration in adult animals (*Espinosa et al., 2001*).

In adult-onset SCA13, an alternative possibility is that slow, intracellular accumulation of the mutant aR3H protein eventually results in neuronal loss due to proteotoxicity, a common mechanism of neurodegeneration in late-onset diseases (*Kurtishi et al., 2019*). The aR3H protein is recognized as abnormal and retained intracellularly in cell lines (*Khare et al., 2017*; *Gallego-Iradi et al., 2014*); intracellular retention of co-assembled wild type subunits is a common mechanism for dominant negative suppression of $K^+$ channel activity (*Schulteis et al., 1998*). A proteotoxic mechanism is consistent with the results of *Namikawa et al., 2019*. In their experiments, Purkinje cells overexpressing the adult-onset mutation died abnormally during cerebellar development. Increasing the level of

expression of the mutant protein accelerated the onset of cell death, which is characteristic of proteotoxicity. Furthermore, Namikawa et al. observed that overexpression of the wild-type Kv3.3 protein, which is not recognized as abnormal and is therefore less likely to accumulate intracellularly, did not trigger Purkinje cell death during cerebellar development (*Namikawa et al., 2019*). In adult-onset SCA13, we suggest that the toxicity of the slowly-accumulated aR3H protein is likely to be more important in triggering cerebellar degeneration than frequency-dependent hypoexcitability.

### Potential roles of altered excitability versus the loss of cerebellar neurons in locomotor deficits in SCA13

Because Kv3.3 is a key regulator of Purkinje cell excitability, it is interesting to consider the relative roles of altered excitability versus loss of cerebellar neurons in the locomotor deficits and other symptoms associated with SCA13. In our experiments, expression of the infant-onset iR4H mutation dramatically altered excitability, which was rapidly followed by Purkinje cell loss. If applicable to human patients, our results would suggest that the early death of cerebellar neurons, potentially triggered by hyperexcitability, is primarily responsible for the motor delay and persistent motor deficits seen in patients with infant-onset SCA13. Expression of the adult-onset aR3H mutation resulted in latent hypoexcitability. A similar reduction in neuronal excitability would be expected in patients throughout life, before the emergence of locomotor symptoms in adulthood. Interestingly, it has been reported that individuals with the aR3H mutation have significant deficits in binaural processing and sound localization before the emergence of motor symptoms (*Middlebrooks et al., 2013*). Kv3.3 is highly expressed in neurons in the medial nucleus of the trapezoid body located in the auditory brainstem (*Grigg et al., 2000*; *Li et al., 2001*). Neuronal hypoexcitabity or subtle morphological changes associated with the adult-onset aR3H mutation may contribute to the auditory processing deficits seen in otherwise pre-symptomatic individuals (*Middlebrooks et al., 2013*). In contrast, the emergence of progressive locomotor deficits is temporally correlated with progressive cerebellar degeneration, suggesting that in adult-onset SCA13, loss of cerebellar neurons underlies the major motor symptoms.

In summary, we have identified differential effects of infant- and adult-onset mutations on Purkinje cell excitability, development, and viability in vivo. Our results are consistent with the existence of two distinct forms of SCA13. The specific correlation of the infant-onset mutation with aberrant maturation and rapid death of Purkinje cells during cerebellar development in vivo indicate that zebrafish is an excellent system for investigating mechanisms that may underlie SCA13 in human patients.

## Materials and methods

**Key resources table**

| Reagent type (species) or resource | Designation | Source or reference | Identifiers | Additional information |
|---|---|---|---|---|
| Gene (*Dario rerio*) | *kcnc3a* | GenBank | HQ118212.1 | |
| Strain, strain background (*Dario rerio*) | Tüpfel long fin nacre | Herwig Baier laboratory hbaier@neuro.mpg.de | | Unpigmented wild type strain |
| Genetic reagent (*Dario rerio*) | la118Tg:*Tg(aldoca:gap43-Venus)* | *Hsieh et al., 2014* | | Transgenic line |
| Recombinant DNA reagent | pBluescript Kv3.3a wild type (plasmid) | *Mock et al., 2010* | | |
| Recombinant DNA reagent | pBluescript Kv3.3a aR3H (plasmid) | *Mock et al., 2010* | | |
| Recombinant DNA reagent | pBluescript Kv3.3a iR4H (plasmid) | This paper | | See Materials and methods, section 2 |
| Recombinant DNA reagent | pminiTol2-aldoca-Kv3.3a wild type-2A-mCherry (plasmid) | This paper | | See Materials and methods, section 2 |

*Continued on next page*

*Continued*

| Reagent type (species) or resource | Designation | Source or reference | Identifiers | Additional information |
|---|---|---|---|---|
| Recombinant DNA reagent | pminiTol2-aldoca-Kv3.3a aR3H-2A-mCherry (plasmid) | This paper | | See Materials and methods, section 2 |
| Recombinant DNA reagent | pminiTol2-aldoca-Kv3.3a iR4H-2A-mCherry (plasmid) | This paper | | See Materials and methods, section 2 |
| Recombinant DNA reagent | pminiTol2-aldoca-mbEGFP (plasmid) | This paper | | See Materials and methods, section 2 |
| Recombinant DNA reagent | pminiTol2-aldoca-Kv3.3a wild type-2A-mbEGFP (plasmid) | This paper | | See Materials and methods, section 2 |
| Recombinant DNA reagent | pminiTol2-aldoca-Kv3.3a aR3H-2A-mbEGFP (plasmid) | This paper | | See Materials and methods, section 2 |
| Recombinant DNA reagent | pminiTol2-aldoca-Kv3.3a iR4H-2A-mbEGFP (plasmid) | This paper | | See Materials and methods, section 2 |
| Recombinant DNA reagent | pminiTol2-aldoca-Kv3.3a iR4H-2A-mbtdTomato (plasmid) | This paper | | See Materials and methods, section 2 |
| Recombinant DNA reagent | pTol2-*gata2* min-(*Mnx1* mne)$_3$-EGFP (plasmid) | *Issa et al., 2012* | | |
| Recombinant DNA reagent | pTol2-*gata2* min-(*Mnx1* mne)$_3$-Kv3.3a aR3H-EGFP fusion (plasmid) | *Issa et al., 2012* | | |
| Recombinant DNA reagent | pTol2-*gata2* min-(*Mnx1* mne)$_3$-Kv3.3a iR4H-EGFP fusion (plasmid) | This paper | | See Materials and methods, section 2 |
| Recombinant DNA reagent | pminiTol2-*gata2* min-(*Mnx1* mne)$_3$-Synaptophysin-mCherry fusion (plasmid) | This paper | | See Materials and methods, section 2 |
| Chemical compound, drug | NS13001 | ChemShuttle, Hayward CA | Cat. #: 104258 | SK channel agonist |
| Chemical compound, drug | Tricaine-S | Syndel, Ferndale WT | MS-222 | Pharmaceutical grade anesthetic |
| Chemical compound, drug | (+)-Tubocurarine hydrochloride pentahydrate | Sigma-Aldrich | T2379 | Paralytic drug |

## Zebrafish maintenance

Zebrafish (*Danio rerio*) were housed in the UCLA Zebrafish Core Facility at 28°C using a 14 hr/10 hr light/dark cycle. Experiments were performed using the unpigmented Tüpfel long fin nacre (TLN) strain or a previously described TLN transgenic line (la118Tg:*Tg(aldoca:gap43-Venus)*) that expresses a membrane-bound form of the Venus yellow fluorescent protein specifically in cerebellar Purkinje cells (*Hsieh et al., 2014*). Adult zebrafish were bred to obtain embryos. The date and time of fertilization were noted for each clutch of embryos to determine the approximate age of an animal at the time of analysis in electrophysiological and imaging experiments. Progeny were raised until 9 days post-fertilization (dpf) in aquarium water in a 28°C incubator using the same light/dark cycle. Starting at 5 dpf, larvae were fed brine shrimp powder twice daily. Zebrafish were euthanized using 0.2% pharmaceutical grade MS-222 (Syndel, Ferndale WA) followed by decapitation. All animal procedures were approved by the Chancellor's Animal Research Committee at UCLA.

## Molecular biology

Bluescript plasmid clones of zebrafish *kcnc3a* cDNAs encoding wild-type Kv3.3 (GenBank Accession #HQ118212.1) and the aR3H mutation were described previously (*Mock et al., 2010*). We have shown that the functional properties of wild-type zebrafish Kv3.3a and the aR3H mutation in zebrafish Kv3.3 strongly resemble the human wild type and mutant proteins (*Minassian et al., 2012*; *Mock et al., 2010*). The iR4H mutation was introduced into the wild-type clone using the Quik-Change mutagenesis kit (Agilent, Santa Clara, CA). As in human Kv3.3, the iR4H mutation in zebrafish Kv3.3 is non-functional when expressed alone, and, when co-expressed with wild type, results in both dominant negative suppression of current amplitude and dominant gain-of-function changes in channel gating depending on the ratio of wild type and mutant subunits in the tetrameric channel (*Figure 1—figure supplement 1*; *Minassian et al., 2012*).

For F0 transgenesis in zebrafish, wild type and mutant Kv3.3 coding sequences were subcloned into the pMiniTol2 vector (Addgene, Waterton MA) using the In-Fusion kit (Takara Bio USA, Mountain View CA) (*Balciunas et al., 2006*). The channel sequence was preceded by the promotor of the zebrafish *aldolase Ca* (*aldoca*) gene, which drives cell-type specific expression in cerebellar Purkinje neurons (*Tanabe et al., 2010*; *Hsieh et al., 2014*). The channel sequence was followed by a viral 2A sequence from porcine teschovirus-1 (*Kim et al., 2011*), and then by sequences encoding mCherry or membrane-bound forms of EGFP (mEGFP) or tdTomato (mTomato) to mark expressing cells. The 2A sequence produces equimolar amounts of wild-type or mutant Kv3.3 and the fluorescent reporter as separate proteins from a single mRNA (*Kim et al., 2011*). Fluorescent proteins mEGFP or mTomato were tethered to the membrane by fusing the first 20 amino acids of zebrafish gap43, which contains a palmitoylation sequence, to the N-terminus (*Tanabe et al., 2010*; *Hsieh et al., 2014*). Plasmid constructs used in control experiments contained the coding sequence for a fluorescent reporter protein, as indicated in the description of the experiment, without the Kv3.3 and 2A sequences.

Alternatively, wild-type or mutant Kv3.3 was specifically expressed in spinal cord motor neurons using previously described Tol2 plasmids encoding zKv3.3a wild type or aR3H with EGFP fused in frame at the C-terminus of the channel protein by a four residue linker (*Issa et al., 2011*; *Issa et al., 2012*). An analogous plasmid was made by introducing the iR4H mutation into the wild-type Kv3.3 sequence using the QuikChange mutagenesis kit (Agilent, Santa Clara, CA). Specific expression in motor neurons was driven by three copies of the 125 bp motor neuron enhancer from the mouse *Mnx1* (*Hb9*) gene (*Nakano et al., 2005*; *Issa et al., 2011*; *Issa et al., 2012*).

A clone encoding a synaptophysin-GFP fusion protein in a UAS vector was the kind gift of Martin Meyer, King's College London (*Meyer and Smith, 2006*). The open reading frame for the fusion protein was transferred to the miniTol2 vector (Addgene, Waterton MA) behind the minimal promotor of the zebrafish *gata2* gene and three copies of the 125 bp motor neuron enhancer from the mouse *Mnx1* (*Hb9*) gene (*Nakano et al., 2005*; *Issa et al., 2011*; *Issa et al., 2012*). The GFP coding sequence was replaced by mCherry using the In-Fusion cloning kit (Takara Bio USA, Mountain View CA).

All plasmid clones were verified by sequencing.

## F0 transgenesis

For mosaic expression of exogenous genes in Purkinje cells or CaP motor neurons, 150–200 pg of plasmid DNA was injected into wild-type TLN embryos at the 1–2 cell stage using a Picospritzer II (Parker Instruments, Hollis NH). Injected embryos were raised in a 28°C incubator with a 14 hr/10 hr light-dark cycle. At ~6 hr post-injection and at 12 hr intervals thereafter, embryos that did not survive or that failed to develop normally were removed and the fish water was replaced. Embryos were manually dechorionated using forceps. Zebrafish were screened for expression in Purkinje cells beginning at three dpf using a Zeiss Discovery V12 epifluorescence microscope (Zeiss, Oberkochen, Germany) or an Olympus Fluoview FV300 laser scanning confocal microscope (Olympus, Tokyo, Japan). Experiments were performed using Purkinje cells in the corpus cerebelli, the lobe of the zebrafish cerebellum with the highest anatomical similarity to the mammalian cerebellum (*Bae et al., 2009*). Zebrafish were screened for expression in CaP motor neurons at ~30 hr post-fertilization (hpf).

## Patch clamp analysis of cerebellar Purkinje cells in living zebrafish

For electrophysiological analysis of Purkinje cell excitability, miniTol2 plasmid DNA encoding the SCA13 mutant aR3H or iR4H subunits or exogenous wild-type Kv3.3a and mCherry under the control of the *aldoca* Purkinje cell-specific promoter was injected into zebrafish embryos of the la118Tg:*Tg (aldoca:gap43-Venus)* line. Uninjected zebrafish of the la118Tg:*Tg(aldoca:gap43-Venus)* line were used in control experiments. Purkinje cells expressing both Venus and mCherry were selected for analysis, or in control experiments, Purkinje cells expressing Venus alone were selected in uninjected fish.

In situ patch clamp analysis of Purkinje cell excitability was performed between 3.5 and 8 dpf as previously described (*Hsieh et al., 2014*). The recording chamber was filled with external solution containing 134 mM NaCl, 2.9 mM KCl, 2.1 mM CaCl$_2$, 1.2 mM MgCl$_2$, 10 mM HEPES, and 10 mM glucose, pH 7.5. Zebrafish were anesthetized with pharmaceutical grade 0.02% MS-222 (Tricaine-S) (Syndel, Ferndale WA) for ~10 s and then glued dorsal side up onto glass coverslips. Zebrafish were paralyzed using 10 µM (+)-tubocurarine hydrochloride pentahydrate dissolved in external solution. Skin around the head and the skull were gently removed using fine forceps (*Hsieh et al., 2014*; *Hsieh and Papazian, 2018*). Zebrafish were transferred to the recording chamber containing external solution. Purkinje cell activity was recorded in awake animals starting 5–10 min after the dissection. Recordings were stable for up to 1 hr. All data presented in this study were acquired within 45 min after the dissection. Zebrafish were euthanized at the end of the experiment.

Data were acquired in the loose patch configuration using an EPC10 patch clamp amplifier and Pulse software (HEKA Elektronik, Holliston MA). Borosilicate pipettes (7–12 MΩ, 1B150F-4 glass) (World Precision Instruments, Sarasota FL) were filled with external solution. Purkinje cells were visualized under an upright Olympus BX51WI microscope using a 40×/0.80 water-immersion lens (Olympus, Tokyo, Japan). Using a motorized micromanipulator, patch pipettes were advanced toward Purkinje cells from the rostral side at an angle of 30° relative to the horizontal plane. Seal resistances ranged from 20 MΩ to 200 MΩ. Experiments were performed at room temperature with the ambient and microscope lights turned off. Electrical activity was recorded in voltage clamp mode at 0 mV. Data were acquired at 20 kHz and filtered at 3 kHz. Complex spikes were identified by visual inspection by the presence of a low amplitude, prolonged depolarization after the initial spike (*Hsieh et al., 2014*).

High frequency firing was evoked in Purkinje cells in response to sudden darkness as previously described (*Hsieh et al., 2014*). Zebrafish in the recording chamber were light adapted for 2 min to a 1W, 6500K white LED light source (Thorlabs, Newton NJ) applied using the halogen light pathway on the microscope. Switching the LED on and off was time-locked with electrophysiological recordings. Purkinje cell activity was recorded for 10 s before and 40 s after the LED was turned off. To avoid over stimulating the visual system, recordings were limited to one or two Purkinje cells per animal. Each Purkinje cell was subjected to 1–4 trials, with an inter-trial interval of at least 2 min. Each 50 s trace was segmented into 500 bins of 100 ms each. The average frequency of firing in each bin was determined and normalized to the average firing frequency calculated over the entire 10 s period before the LED was turned off at time 0 s. The fold-change in frequency from all trials was then averaged and plotted versus time.

Where indicated, 10 or 20 µM NS13001 (ChemShuttle, Hayward CA), an activator of small conductance, Ca$^{2+}$-activated K$^+$ (SK) channels (*Kasumu et al., 2012*), was added to external solution by diluting from a 20 mM stock solution in DMSO.

Electrophysiological data were analyzed using Clampfit 10 (Molecular Devices, San Jose CA). Data acquired with the EPC10 amplifier (HEKA Elektronik, Holliston MA) were imported into Igor 6.2 (WaveMetrics, Tigard OR) and converted to a Clampfit-compatible format for analysis. The regularity of spontaneous tonic firing was quantified by determining the coefficient of variation of adjacent intervals (CV2) defined as $\frac{2}{n}\sum_{1}^{n}\frac{|I_{i+1}-I_i|}{I_{i+1}+I_i}$, where I is the interspike interval in ms. CV2 is preferable to the conventional coefficient of variation (CV, standard deviation/mean) for data recorded on a short time scale (*Walter et al., 2006*; *Wulff et al., 2009*). Data are provided as mean ± SEM. Statistical significance was assessed using Excel (Microsoft, Seattle WA) or Origin 8 (OriginLab, Northampton MA) software. Non-firing cells were not included in the analysis because it was not feasible to

differentiate between cells that were healthy and silent versus cells that were damaged during the dissection or patch formation.

In box plots, the box indicates the 25th to 75th percentile range of the data, the whiskers indicate the 5th to 95th percentile range, the open square symbol indicates the mean value, and the solid horizontal line across the box indicates the median.

## Live confocal imaging of Purkinje cells

MiniTol2 plasmid DNA encoding the aR3H or iR4H mutant subunits or exogenous wild-type Kv3.3a and mEGFP, or mEGFP alone under the control of the *aldoca* promotor was injected into TLN embryos. Purkinje cell development and viability were assessed by repeated confocal imaging in live zebrafish between 3 and 8 dpf. Isolated Purkinje cells expressing mEGFP or traceable cells in small groups of 2–3 expressing cells were selected for analysis starting at 3 dpf. Zebrafish were anesthetized with pharmaceutical grade 0.02% MS-222 (Tricaine-S) (Syndel, Ferndale WA) and embedded dorsal side up in 1% low melt ultra-pure agarose (ThermoFisher Scientific, Waltham MA). Images were acquired as 1 µm optical sections using an Olympus Fluoview FV300 laser scanning confocal microscope (Olympus, Tokyo, Japan). The same cells or small groups of cells were imaged on consecutive days. Zebrafish were carefully removed from the agarose between time points and re-embedded for subsequent imaging.

Single expressing Purkinje cells were traced from confocal image stacks for three-dimensional digital reconstruction using Imaris 8.0 (Bitplane, Concord MA). Total process length and the number of branches for each cell were obtained from the traced images. Spine number was counted manually using ImageJ (NIH, Bethesda MD). Spines in each cell were counted three times and the values, which differed by less than 10%, were averaged. Statistical analyses were performed using SPSS software (IBM, Armonk NY). Values were $\log_{10}$-transformed because individual data points were right-skewed rather than normally distributed. Comparisons were made using a Linear Mixed Model with Bonferroni post-hoc tests. Statistical significance was assessed as $p < 0.05$. Figures were prepared using Adobe Illustrator (Adobe, San Jose CA).

## Chronic NS13001 treatment

Fertilized zebrafish embryos were randomly divided into two groups. At the 1–2 cell stage, one group of embryos was injected with a mini-Tol2 plasmid encoding iR4H and mEGFP separated by a 2A sequence (*Kim et al., 2011*), whereas the other group was injected with an equimolar amount of a plasmid encoding mEGFP alone. Purkinje cell-specific expression was driven by the *aldoca* promoter. Chronic treatment of zebrafish with NS13001 was initiated at 3.25 dpf. NS13001 was diluted into fish water to a final concentration of 20 µM using a 20 mM stock solution in DMSO. In control experiments, an equivalent amount of DMSO was added to fish water. Live confocal imaging commenced at 4 dpf and was repeated at 24 hr intervals. The fish water was changed and the drug or DMSO was refreshed after each imaging session. The total number of expressing Purkinje cells in the cerebellum was counted manually in each zebrafish using ImageJ (NIH, Bethesda MD).

## Acridine orange staining

Acridine orange is a fluorescent molecule that enters living and early apoptotic cells, intercalates into DNA, and stains the nucleus green. In the presence of condensed and fragmented chromatin, a hallmark of early apoptotic cells, bright green puncta are observed (*Atale et al., 2014*). To detect Purkinje cell death using acridine orange, a miniTol2 plasmid DNA encoding the iR4H mutant subunit and membrane-bound Td-Tomato (mTomato) under the control of the *aldoca* promotor was injected into TLN embryos. mTomato was used instead of mEGFP to mark expressing cells because acridine orange fluoresces green (*Atale et al., 2014*). Purkinje cells expressing iR4H and mTomato were imaged in live zebrafish at 4 dpf using an Olympus Fluoview FV300 laser scanning confocal microscope (Olympus, Tokyo, Japan). At 5 dpf, zebrafish were incubated in 1.5 µl/ml acridine orange (Invitrogen, Carlsbad CA) for 1 hr at 28°C. Zebrafish were then washed three times for 5 min in fish water lacking acridine orange. The staining and washing protocol was performed under low light to minimize bleaching of acridine orange. Zebrafish were imaged within 15 min of the final wash and then euthanized.

## Live confocal imaging of CaP motor neurons

MiniTol2 plasmid DNA encoding fusion proteins of the aR3H or iR4H mutant subunits with EGFP, or EGFP alone, was mixed with DNA encoding synaptophysin-mCherry, a marker of presynaptic vesicles, at a 1:1 ratio (*Meyer and Smith, 2006*; *Issa et al., 2011*; *Issa et al., 2012*). DNA (~180 pg total per embryo) was injected into TLN embryos at the 1–2 cell stage (*Issa et al., 2011*). Expression was driven by the motor neuron enhancer from the mouse *Mnx1* (*Hb9*) gene (*Nakano et al., 2005*). Injected animals were screened for EGFP fluorescence at ~30 hpf using a Zeiss Discovery V12 epi-fluorescence microscope (Zeiss, Oberkochen, Germany). At 48 hpf, expressing animals were anesthetized with 0.02% MS-222 in embryo water and embedded in 0.8% ultrapure low melt agarose (ThermoFisher Scientific, Waltham MA). The agarose was then covered with embryo water containing 0.02% MS-222. Confocal imaging was performed on live, morphologically normal animals as previously described (*Issa et al., 2011*; *Issa et al., 2012*). Images were acquired as 1 µm optical sections at ~48 hpf (48–52 hfp) hpf using an Olympus FV300 Fluoview laser scanning confocal microscope equipped with a LUMPLFL 40_0.8 NA water immersion lens (Olympus, Tokyo Japan).

Confocal images and image aspect ratio parameters were imported into Imaris 8.0 (Bitplane, Concord MA) for digital tracing and three-dimensional reconstruction. Analysis was restricted to neurons located in somites 10–18 so that the analyzed population would be similar in size and developmental stage. Quantified parameters of traced neurons were exported into Excel. Statistical analysis was performed using GraphPad Prism software v.3.03.

## Acknowledgements

We are grateful to Drs. Tom Otis, Joanna Jen, and Bal Khakh for helpful advice. We thank Allan Mock, Eoon Hye, and Emily Chang for excellent technical assistance and Martin Meyer, King's College London, for kindly providing a plasmid clone of a synaptophysin-GFP fusion protein. We are grateful to the UCLA Department of Biostatistics Consultation Service for advice on statistical analysis.

## Additional information

### Funding

| Funder | Grant reference number | Author |
|---|---|---|
| National Institutes of Health | R01 NS058500 | Diane M Papazian |
| National Ataxia Foundation | | Fadi A Issa |
| UCLA Stein Oppenheimer Seed Grant | | Diane M Papazian |
| UCLA Jennifer Buchwald Graduate Fellowship | | Jui-Yi Hsieh Brittany N Ulrich |

The funders had no role in study design, data collection and interpretation, or the decision to submit the work for publication.

### Author contributions

Jui-Yi Hsieh, Conceptualization, Formal analysis, Investigation, Methodology, Writing - review and editing; Brittany N Ulrich, Fadi A Issa, Meng-chin A Lin, Formal analysis, Investigation, Methodology, Writing - review and editing; Brandon Brown, Formal analysis, Investigation; Diane M Papazian, Conceptualization, Formal analysis, Supervision, Funding acquisition, Investigation, Methodology, Writing - original draft, Writing - review and editing

### Author ORCIDs

Fadi A Issa  http://orcid.org/0000-0001-5234-5850
Diane M Papazian  https://orcid.org/0000-0001-8194-5740

## Ethics

Animal experimentation: This study was performed in strict accordance with the recommendations in the Guide for the Care and Use of Laboratory Animals of the National Institutes of Health. All of the animals were handled according to approved institutional animal care and use committee (IACUC) protocols of the University of California, Los Angeles. The protocols were approved by the Chancellor's Animal Research Committee (#2005-176 and #1991-329). All surgery was performed under MS-222 anesthesia, and every effort was made to minimize suffering.

### Decision letter and Author response

Decision letter https://doi.org/10.7554/eLife.57358.sa1
Author response https://doi.org/10.7554/eLife.57358.sa2

# Additional files

### Supplementary files

• Transparent reporting form

### Data availability

All data generated or analyzed during this study are included in the manuscript and supporting files.

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

## Appendix 1

### Supplemental Methods

### Expression of Kv3.3 in Xenopus oocytes and voltage clamp analysis

Oocytes were obtained from *Xenopus laevis* frogs using a protocol approved by the Chancellor's Animal Research Committee at UCLA. Bluescript plasmid clones of zebrafish *kcnc3a* cDNA encoding wild-type Kv3.3a or the iR4H mutation were linearized and transcribed using the mMessage mMachine T7 Ultra kit (Invitrogen, Waltham MA) (*Mock et al., 2010*). RNA was injected into *Xenopus* oocytes, which were incubated at 18 ± 2°C for 1–3 days before electrophysiological experiments (*Timpe et al., 1988*; *Papazian et al., 1991*). Channel function was analyzed using a Warner OC-725C two-electrode voltage clamp (Warner Instruments, Hamden CT) (*Mock et al., 2010*; *Minassian et al., 2012*). Ionic currents were recorded at room temperature (18 ± 2°C). Electrodes were filled with 3 M KCl and had resistances of 0.3–1.0 MΩ. The bath solution contained 85 mM NaCl, 4 mM KCl, 1.8 mM $CaCl_2$, and 10 mM HEPES, pH 7.2. Pulse protocols were generated and data were acquired using pClamp software (Molecular Devices, San Jose CA). Data were sampled at 10 kHz and filtered at 2 kHz. Linear capacitive and leak currents were subtracted using a p/-4 protocol.

The voltage dependence of channel opening was characterized by pulsing the membrane from a holding potential of −90 mV to voltages ranging from −40 to +70 mV in 10 mV increments. Conductance values were calculated from peak current amplitudes assuming a linear open channel current-voltage relationship and a reversal potential of −80 mV. Conductance values were normalized to the maximum value obtained in the experiment and plotted versus voltage. Data were fitted with a single Boltzmann function to estimate the midpoint voltage, $V_{0.5}$, and slope factor.

