## [Decision Letter]

**Acceptance summary:**

This manuscript tackles the underlying differences between two human spinocerebellar ataxia 13 mutations, one severe and appearing in infancy, and the other with a milder late adult onset. The Kv3.3 channel mutations are closely related, in that each is an R to H mutation in one of the positive charges of the voltage-sensing domain. Here the authors address this problem in zebrafish Purkinje cells, and find striking differences between the two mutants. The work demonstrates that relatively modest changes in channel properties can balloon into major alterations of cellular activity, circuit function and even viability. It also reveals that mutations even in well-studied channel domains cannot easily be judged simply by eye and intuition.

**Decision letter after peer review:**

Thank you for submitting your article "Infant and adult SCA13 mutations differentially affect Purkinje cell excitability, maturation, and viability in vivo" for consideration by *eLife*. Your article has been reviewed by three peer reviewers, including Julie A Kauer as the Reviewing Editor and Reviewer #1, and the evaluation has been overseen by Richard Aldrich as the Senior Editor.

The reviewers have discussed the reviews with one another and the Reviewing Editor has drafted this decision to help you prepare a revised submission.

Summary:

In this work, Kv3.3 channels expressing two closely spaced mutations that in humans lead to either infant onset or later adult onset spinocerebellar ataxia were introduced into Purkinje cells of larval zebrafish. The early-onset mutation generated dramatic increases in Purkinje cell firing, disruptions of morphological development, and cell death, consistent with the primarily gain-of-function biophysical phenotype that the authors previously found in vitro. Instead, the late-onset mutation led only to a moderate reduction in excitability, consistent with the loss-of-function in vitro phenotype. The work demonstrates that fairly modest changes in biophysical properties as assayed in vitro can balloon into major alterations of cellular activity, circuit function, and even viability, in a way that is relevant to basic biology as well as disease states. It also reveals that mutations even in well-studied domains of a channel cannot by judged simply as likely to increase or decrease excitability, or promote a specific circuit dysfunction.

Essential revisions:

The reviewers were enthusiastic about the manuscript, but felt it would benefit from a more thorough discussion of limitations/likely interpretations. The authors were very frank about drawing our attention to the problematic parts, where everything isn't perfectly neat in design or outcomes (e.g., fish dying in the rescue drug). But they should go a step further and call out the limitations, as well as what we can learn despite the limitations, with justification. For example, in the rescue experiments, the authors should be explicit about what the discovery is here. It isn't that firing is slowed; it is that viability is increased, providing a possible mechanistic link between the elevated firing and cell death. It also helps separate the pathways that leads to morphological changes (unaffected by the drug) from those that lead to death (affected by the drug). This needs to be spelled out clearly and logically, and what conclusions can reasonably be drawn in their opinions.

1) Subsection “Infant-onset iR4H mutation transiently increases Purkinje cell excitability during cerebellar development in vivo”, second paragraph, last 2 sentences: Figure 1—figure supplement 1 presents data for the iR4H variant only. Also, it is important to clarify that the data refer to work in a heterologous expression system and not in vivo. In this regard, the data of Figure 2—figure supplement 3 are of interest. Is it possible that regardless of whether the transgenically expressed subunits are functional or not, neurons may limit the amount of Kv3.3 conductance? None the less, the data of Figure 2—figure supplement 3 help interpret the results obtained by transgenic expression of the mutant kv3.3a subunits. Are the control data of Figure 2—figure supplement 3 the same as the control 5.25 dpf data of Figure 1?

2) Figure 6: It is hard to know how relevant the effects of transgenic expression of kv3.3a mutants on presynaptic development of CaP neurons informs about potential effects on cerebellar Purkinje cells. For one, the CaP synapses occur in the periphery while those of the Purkinje cell would be within the central nervous system. CaPs and Purkinje cells could use different mechanisms for presynaptic development. Unless the authors can provide a strong argument that effects on CaPs are relevant to Purkinje cells on the basis of prior work and the literature, these data seem to distract from what is otherwise an appropriate focus of the manuscript, seem out of place as presented. They do provide a point of contrast that keeps one from thinking that everything is happening at the level of intrinsic firing properties of Purkinje cells. The authors need to make it make sense to the reader if they want it in there. Please discuss thoughts on how the authors believe these findings fit in, and move the data to supplementary information.

3) Results subsection “Infant-onset iR4H mutation transiently increases Purkinje cell excitability during cerebellar development in vivo” and Figure 1B. Can the authors comment on whether the observation that complex spikes are greatly reduced in the iR4H mutant Purkinje cells stems from the deterioration of the climbing fiber synaptic contacts or an actual lack of firing by the inferior olive? The data may not fully address this question, which is fine, but it is mentioned in Figure 1 and is very interesting, so some return and discussion of this question would be informative.

4) Figure 3C, please explain how the firing frequency and fold-change were calculated. The legend says that the frequency was measured in 100 ms intervals. How does this work when the mean firing rate is less than 10 Hz, i.e., < 1 spike per interval? Is this a summed number of spikes per 100 ms bin over many trials (how many?) converted to a rate? Is the plotted data in 3C the illustration for a single cell or is that an average? If the former, can population data be shown? If the latter, can some indication of the spread of data be shown?

5) An apparently unconsidered point in the Discussion is whether all the changes in excitability and sequelae are directly due to the changes in potassium channel function. Many studies have documented secondary changes (often but not always compensatory) when one channel is disrupted. Specifically in Purkinje cells, changes in sodium channel current led to changes in potassium currents (Khaliq, Gouwens, and Raman, 2003). This idea seems important to acknowledge, since not all the changes in excitability necessarily stem exclusively from the altered Kv3.3 properties, and certainly other molecular changes must take place to account for the morphological changes and cell death.

---

## [Author Response]

Essential revisions:The reviewers were enthusiastic about the manuscript, but felt it would benefit from a more thorough discussion of limitations/likely interpretations. The authors were very frank about drawing our attention to the problematic parts, where everything isn't perfectly neat in design or outcomes (e.g., fish dying in the rescue drug). But they should go a step further and call out the limitations, as well as what we can learn despite the limitations, with justification. For example, in the rescue experiments, the authors should be explicit about what the discovery is here. It isn't that firing is slowed; it is that viability is increased, providing a possible mechanistic link between the elevated firing and cell death. It also helps separate the pathways that leads to morphological changes (unaffected by the drug) from those that lead to death (affected by the drug). This needs to be spelled out clearly and logically, and what conclusions can reasonably be drawn in their opinions.

We made several revisions in the Discussion, under the heading “Different pathogenic mechanisms likely contribute to infant- and adult-onset SCA13”, to address this point.

1) In the first paragraph in this section, we added a sentence to more completely describe the timeline of events in our experiments: “Large increases in spike frequency by 4.25 dpf occurred concurrently with significant disruption of Purkinje cell development.”

2) Later in that same paragraph, we clarify what has been learned: “Consistent with the proposal that hyperexcitability is mechanistically linked to Purkinje cell loss, suppressing firing with the SK channel activator NS13001 increased the early survival of iR4H-expressing neurons. In contrast, treatment with NS13001 did not restore normal morphological maturation, suggesting that different pathways contribute to aberrant development and cell death.”

3) The second paragraph has been revised to clarify that changes in Ca^2+^ handling could activate different pathways that affect Purkinje cell development and viability: “Because it regulates multiple signaling pathways, Ca^2+^ is a plausible intermediate between hyperexcitability and the maldevelopment and death of iR4H-expressing Purkinje cells. High frequency firing is expected to increase Ca^2+^ entry, which in turn may induce Ca^2+^-activated Ca^2+^ release from internal stores...”

4) The third paragraph flags caveats that apply to the rescue experiments, starting: “Several caveats apply to the NS13001 experiments.”

1) Subsection “Infant-onset iR4H mutation transiently increases Purkinje cell excitability during cerebellar development in vivo”, second paragraph, last 2 sentences: Figure 1—figure supplement 1 presents data for the iR4H variant only. Also, it is important to clarify that the data refer to work in a heterologous expression system and not in vivo.

The Results section under the heading “Infant-onset iR4H mutation transiently increases Purkinje cell excitability during cerebellar development in vivo” has been revised to read: “It is important to note that the iR4H and aR3H subunits are non-functional in the absence of wild-type Kv3.3 subunits. […] Accordingly, these mutant subunits would have to co-assemble with endogenous, wild-type Kv3 subunits to affect Purkinje cell excitability.”

In this regard, the data of Figure 2—figure supplement 3 are of interest. Is it possible that regardless of whether the transgenically expressed subunits are functional or not, neurons may limit the amount of Kv3.3 conductance? None the less, the data of Figure 2—figure supplement 3 help interpret the results obtained by transgenic expression of the mutant kv3.3a subunits.

Previous workers have compared the total Kv3 current amplitude in mammalian Purkinje cells with the amplitude of the current activated during an action potential, revealing that a minority of channels open during a spike. If the same is true in zebrafish Purkinje cells, it may provide a plausible explanation for our results. To address this point, we added a new paragraph to the Discussion under the heading “Infant- and adult-onset SCA13 mutations have opposite effects on excitability and differentially affect Purkinje cell maturation and survival during cerebellar development in vivo”: “Taken together, our results indicate that increasing or decreasing Kv3.3 activity by expressing exogenous wild type or dominant negative aR3H subunits, respectively, does not alter basal excitability in Purkinje cells. […] During basal tonic firing, a similar spare capacity in zebrafish Purkinje cells could suffice to buffer the spike frequency against the changes in Kv3 current amplitude that occur in our experiments.”

Are the control data of Figure 2—figure supplement 3 the same as the control 5.25 dpf data of Figure 1?

Yes, this is stated in the legend to part B of Figure 2—figure supplement 3: “Control data obtained at 5.25 dpf are the same as shown in Figure 1C and are repeated here for comparison to exoWT.” We added the following sentence to the legend for part C of Figure 2—figure supplement 3: “Control data obtained at 5.25 dpf are the same as shown in Figure 2—figure supplement 2C and are repeated here for comparison to exoWT.”

2) Figure 6: It is hard to know how relevant the effects of transgenic expression of kv3.3a mutants on presynaptic development of CaP neurons informs about potential effects on cerebellar Purkinje cells. For one, the CaP synapses occur in the periphery while those of the Purkinje cell would be within the central nervous system. CaPs and Purkinje cells could use different mechanisms for presynaptic development. Unless the authors can provide a strong argument that effects on CaPs are relevant to Purkinje cells on the basis of prior work and the literature, these data seem to distract from what is otherwise an appropriate focus of the manuscript, seem out of place as presented. They do provide a point of contrast that keeps one from thinking that everything is happening at the level of intrinsic firing properties of Purkinje cells. The authors need to make it make sense to the reader if they want it in there. Please discuss thoughts on how the authors believe these findings fit in, and move the data to supplementary information.

To place these results in context, the Results section headed “Infant-onset but not adult-onset mutation disrupts presynaptic development” has been extensively revised. The section now starts with a brief description of relevant results from earlier publications. The relevance of motor neurons to this study is mentioned. The former Figure 6 has been moved to the supplementary section as the new Figure 5—figure supplement 1.

3) Results subsection “Infant-onset iR4H mutation transiently increases Purkinje cell excitability during cerebellar development in vivo” and Figure 1B. Can the authors comment on whether the observation that complex spikes are greatly reduced in the iR4H mutant Purkinje cells stems from the deterioration of the climbing fiber synaptic contacts or an actual lack of firing by the inferior olive? The data may not fully address this question, which is fine, but it is mentioned in Figure 1 and is very interesting, so some return and discussion of this question would be informative.

Our results do not address the state of synaptic endings in climbing fibers or the firing of inferior olive neurons, so we have refrained from speculating about these issues. To address the reviewers’ concern, we have highlighted two results that likely contribute to the lack of complex spiking in iR4H-expressing Purkinje cells. First, (in response to a comment below), we included the fraction of control and iR4H-expressing cells that exhibit complex spiking at different times post-fertilization (subsection “Infant-onset iR4H mutation transiently increases Purkinje cell excitability during cerebellar development in vivo”). Most control cells do not show complex spiking until 5.25 dpf, by which time the vast majority of iR4H-expressing cells are electrically inexcitable, which likely precludes complex spiking. Under the heading “Infant-onset iR4H mutation transiently increases Purkinje cell excitability during cerebellar development in vivo”, we added the following sentence: “No complex spiking was observed at 5.25 dpf, in accord with the observation that most iR4H-expressing Purkinje cells were electrically silent by that time.” Second, under the heading “Infantonset iR4H mutation disrupts Purkinje cell development”, the lack of dendritic spines in iR4H-expressing cells is described. We added a new sentence indicating that the lack of postsynaptic development may disrupt complex spiking: “The relative lack of dendritic spines may contribute to impaired complex spiking in iR4H-expressing Purkinje cells.”

4) Figure 3C, please explain how the firing frequency and fold-change were calculated. The legend says that the frequency was measured in 100 ms intervals. How does this work when the mean firing rate is less than 10 Hz, i.e., < 1 spike per interval? Is this a summed number of spikes per 100 ms bin over many trials (how many?) converted to a rate? Is the plotted data in Figure 3C the illustration for a single cell or is that an average? If the former, can population data be shown? If the latter, can some indication of the spread of data be shown?

We added the following text to the Materials and methods section under the heading “Patch clamp analysis of cerebellar Purkinje cells in living zebrafish”: “Each 50 s trace was segmented into 500 bins of 100 ms each. […] The fold-change in frequency from all trials was then averaged and plotted versus time.” In addition, the legend to Figure 3C has been revised to make it clearer that the data from all the cells in each group have been averaged. Gray shading has been included in the panel to illustrate the SEM, providing an indication of the spread of the data.

5) An apparently unconsidered point in the Discussion is whether all the changes in excitability and sequelae are directly due to the changes in potassium channel function. Many studies have documented secondary changes (often but not always compensatory) when one channel is disrupted. Specifically in Purkinje cells, changes in sodium channel current led to changes in potassium currents (Khaliq, Gouwens, and Raman, 2003). This idea seems important to acknowledge, since not all the changes in excitability necessarily stem exclusively from the altered Kv3.3 properties, and certainly other molecular changes must take place to account for the morphological changes and cell death.

In the Discussion section under the heading “Changes in Purkinje cell excitability are consistent with effects of infant- and adult-onset mutations on Kv3.3 function”, we added the following sentence: “Additionally, compensatory changes in the expression of other channels may modulate the firing properties that we have observed (Khaliq et al., 2003).”